# Sources of variation in the serum metabolome of female participants of the HUNT2 study
Julia Debik [1,8] ✉, Katarzyna Mrowiec[2,8], Agata Kurczyk [3], Piotr Widłak[4], Karol Jelonek [2], Tone F. Bathen[5,6] & Guro F. Giskeødegård [1,7] ✉

The aim of this study was to explore the intricate relationship between serum metabolomics and lifestyle factors, shedding light on their impact on health in the context of breast cancer risk. Detailed metabolic profiles of 2283 female participants in the Trøndelag Health Study (HUNT study) were obtained through nuclear magnetic resonance (NMR) spectroscopy and mass spectrometry (MS). We show that lifestyle-related variables can explain up to 30% of the variance in individual metabolites. Age and obesity were the primary factors affecting the serum metabolic profile, both associated with increased levels of triglyceride-rich very low-density lipoproteins (VLDL) and intermediate-density lipoproteins (IDL), amino acids and glycolysis-related metabolites, and decreased levels of high-density lipoproteins (HDL). Moreover, factors like hormonal changes associated with menstruation and contraceptive use or education level influence the metabolite levels.

Participants were clustered into three distinct clusters based on lifestyle-related factors, revealing metabolic similarities between obese and older individuals, despite diverse lifestyle factors, suggesting accelerated metabolic aging with obesity. Our results show that metabolic associations to cancer risk may partly be explained by modifiable lifestyle factors.

Circulating metabolic profiles have repeatedly been found to be associated with a wide range of disease outcomes in large-scale population studies[1-4]. Biofluids represent an easily accessible, minimally-invasive material suitable for disease detection and monitoring. While urine is a biological "waste" material containing metabolic breakdown products from various sources, such as food intake or medications, serum and plasma may better reflect interactions between lifestyle factors and endogenous processes driven by the proceeding omics levels. Studying the serum or plasma metabolome gives us a detailed systemic picture of the current state of an organism.

It is well-known that the circulating metabolic profile is dependent on both internal and external factors, and the high inter-individual variation and heterogeneity across different cohorts may be the main reason why the clinical translation of metabolic biomarkers from large-scale epidemiological studies is challenging. The abundance of some metabolites are partially explained by the gut microbiome[5,6], by the consumption of specific foods[7,8],

body weight[9,10], metabolic diseases[11-13], heritability[14], and pre-analytical sampling procedures[15-18]. Nevertheless, the primary factors influencing the levels of most metabolites are unknown.

Breast cancer is the most common cancer disease among women worldwide, and even though the prognosis has been much improved in the last decades, the incidence rate continues to increase[19]. Preventive measures are thus crucial to reduce the cancer burden. There are several established breast cancer risk factors including postmenopausal obesity, high alcohol consumption, reproductive history, and physical inactivity. A high consumption of fruits, vegetables, soy, and fish[20] has been associated with a decreased breast cancer risk, while a western diet has been associated with an increased risk[21]. Metabolic syndrome has been strongly associated with increased breast cancer risk[22]. The characterization of metabolites, lipoproteins, and lipids associated with breast cancer risk factors may provide insights into mechanisms leading to breast cancer development, and

[1]Department of Public Health and Nursing, Norwegian University of Science and Technology, Trondheim, Norway. [2]Center for Translational Research and Molecular Biology of Cancer, Maria Skłodowska-Curie National Research Institute of Oncology, Gliwice Branch, Gliwice, Poland. [3]Department of Biostatistics and Bioinformatics, Maria Skłodowska-Curie National Research Institute of Oncology, Gliwice Branch, Gliwice, Poland. [4]2nd Radiology Department, Medical University of Gdańsk, Gdańsk, Poland. [5]Department of Circulation and Medical Imaging, Norwegian University of Science and Technology, Trondheim, Norway. [6]Department of Radiology and Nuclear Medicine, St. Olav's University Hospital, Trondheim, Norway. [7]Clinic of Surgery, St. Olav's University Hospital, Trondheim, Norway. [8]These authors contributed equally: Julia Debik, Katarzyna Mrowiec. ✉e-mail: julia.b.debik@ntnu.no; guro.giskeodegard@ntnu.no

potentially contribute to identifying preventive measures. We have previously shown associations between serum metabolic markers and long-term risk of breast cancer[23,24]. Furthermore, a study investigating associations between metabolic measures and cancer preventive measures, showed that behaviors related to obesity, and fruit, vegetable, or alcohol consumption had a large metabolic impact[25]. It has also been shown that maintaining a healthy metabolome is beneficial for reducing the risk of developing cardiovascular disease especially for obese individuals[9], and that adherence to healthy lifestyle factors also improves metabolic profiles in diabetic individuals[26]. Moreover, the effect of increased levels of cardiorespiratory fitness on lipids and lipoproteins has been investigated[27], where a study in men showed that dyslipidemia and related diseases may be delayed by maintaining a favorable lipid and lipoprotein profile through improved cardiorespiratory fitness[28]. We thus hypothesize that breast cancer prevention may to some extent be accomplished by maintaining a healthy metabolic profile.

In this paper, we aimed to investigate the main sources of variation, considering both breast cancer-related and other lifestyle factors, in the female serum metabolome, offering valuable insight into metabolic pathways that might be relevant for breast cancer prevention. Breast-cancer related factors considered in this study are alcohol consumption, menarche age, height, age at first pregnancy, number of full-term pregnancies, obesity, physical activity, systemic menopausal estrogen use and birth control pill use[29].

## Results
### Study population

In the current analysis, 2283 female participants aged 20–91 years, who had donated blood samples within the Trøndelag Health Study (HUNT2 study) in the years 1995–1997 (Fig. 1) were included. Nuclear magnetic resonance (NMR)-measured metabolic data was available for the full cohort, while mass spectrometry (MS) data was available for a subcohort of 815 women (Methods Section, Fig. 1). The metabolic profiles in this study thus include the levels of in total 89 NMR measured lipoprotein subfractions and 28 metabolites, and 254 lipids and 30 metabolites (amino acids and biogenic amines) measured by MS. Figure 1 shows a flow-chart of the study setup. After data cleansing and eliminating lifestyle variables with more than 30% missing values, 60 variables related to demography, lifestyle, and socio-economic factors, and anthropometric and clinical measurements (lifestyle-related factors in short) were retained for statistical analyses (Figure S1, Supplementary Data 1). The characteristics of the study population are shown in Table 1. The median age at participation in HUNT2 was 51 years (IQR: 43–62 years); median diastolic and systolic blood pressures were 79 and 132 mmHg, respectively (IQR: 71–87 and 119–150 mmHg); median body mass index (BMI) was 25.8 kg/m$^2$ (IQR: 23.2–29.0 kg/m$^2$); 69% of the participants were married; 57% had good self-reported health; 45% had lower education; and 63% were either inactive or below the recommended physical activity level.

**Fig. 1 | Flow chart summarizing the data flow in this study.** Lifestyle variables were obtained from the HUNT2 databank, and include variables related to clinical parameters, demography, lifestyle, socio-economic factors, and anthropometric measurements. Blood samples were collected in the years 1995-97 and were stored in the HUNT biobank until metabolic profiling by NMR and MS in 2019. N and v indicate the number of individuals and number of variables at each step, respectively. NMR: nuclear magnetic resonance; MS: mass spectrometry.

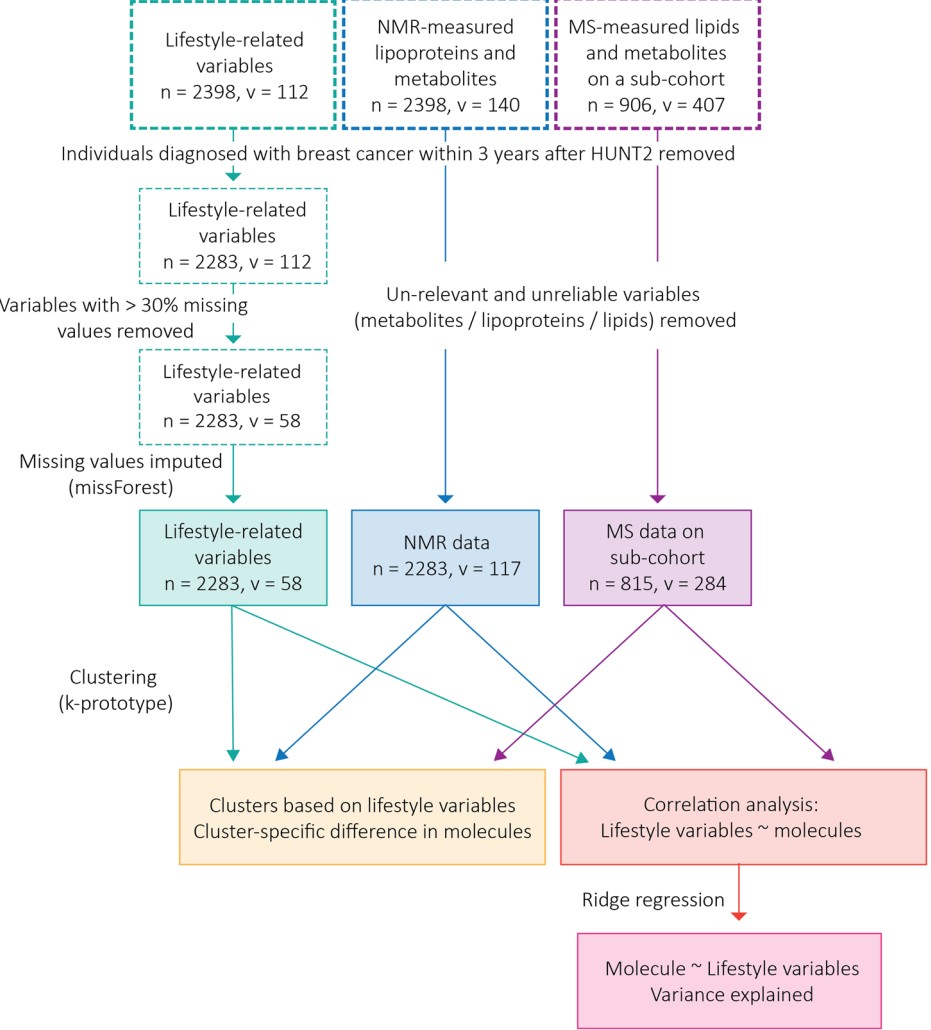

**Table 1 | Baseline characteristics of the study cohort**

| Lifestyle variable | Short name | Overall, N = 2,283[a] | Cluster 1, N = 494[a] | Cluster 2, N = 1,236[a] | Cluster 3, N = 553[a] | P-value[b] |
|---|---|---|---|---|---|---|
| Breast cancer incidence | BC | 1084 (47%) | 237 (48%) | 601 (49%) | 246 (44%) | 0.266 |
| Mean diastolic blood pressure [mmHg] | BPDiasMn23 | 79 (71, 87) | 83 (76, 90) | 74 (68, 80) | 88 (80, 96) | <0.001 |
| Mean systolic blood pressure [mmHg] | BPSystMn23 | 132 (119, 150) | 142 (131, 155) | 122 (114, 131) | 157 (145, 173) | <0.001 |
| Height [cm] | Height | 164 (160, 168) | 164 (160, 169) | 166 (162, 170) | 160 (156, 164) | <0.001 |
| Weight [kg] | Weight | 70 (63, 78) | 87 (81, 93) | 66 (61, 72) | 69 (63, 74) | <0.001 |
| Waist circumference [cm] | WaistCirc | 80 (74, 89) | 96 (92, 101) | 75 (71, 80) | 83 (78, 88) | <0.001 |
| Hip circumference [cm] | HipCirc | 101 (96, 108) | 113 (109, 118) | 98 (94, 102) | 102 (98, 106) | <0.001 |
| Body mass index [kg/m$^2$] | BMI | 25.8 (23.2, 29.0) | 32.0 (29.7, 34.5) | 23.9 (22.2, 25.9) | 26.8 (24.7, 28.6) | <0.001 |
| Waist-to-hip ratio | WHR | 0.80 (0.76, 0.84) | 0.85 (0.81, 0.89) | 0.77 (0.74, 0.80) | 0.81 (0.78, 0.85) | <0.001 |
| Participation age [y] | PartAg | 51 (43, 62) | 55 (48, 64) | 45 (38, 51) | 68 (58, 75) | <0.001 |
| Health | Health | | | | | <0.001 |
| Poor | | 31 (1.4%) | 15 (3.0%) | 8 (0.6%) | 8 (1.4%) | |
| Not so good | | 668 (29%) | 218 (44%) | 255 (21%) | 195 (35%) | |
| Good | | 1,301 (57%) | 239 (48%) | 743 (60%) | 319 (58%) | |
| Very good | | 283 (12%) | 22 (4.5%) | 230 (19%) | 31 (5.6%) | |
| Chronic disease | DisChr | 823 (36%) | 247 (50%) | 302 (24%) | 274 (50%) | <0.001 |
| Alcohol frequency monthly | Alcohol | 1.00 (0.00, 2.17) | 0.00 (0.00, 2.00) | 2.00 (0.00, 3.00) | 0.00 (0.00, 1.22) | <0.001 |
| Smoking | Smoking | 4.0 (0.0, 10.0) | 4.1 (0.0, 10.0) | 6.0 (0.0, 10.0) | 0.0 (0.0, 5.0) | <0.001 |
| Physical activity level | PhysAct | | | | | <0.001 |
| Inactive | | 602 (26%) | 202 (41%) | 208 (17%) | 192 (35%) | |
| Below recommended | | 821 (36%) | 174 (35%) | 439 (36%) | 208 (38%) | |
| Recommended | | 614 (27%) | 93 (19%) | 392 (32%) | 129 (23%) | |
| Above recommended | | 246 (11%) | 25 (5.1%) | 197 (16%) | 24 (4.3%) | |
| No. of full-term pregnancies | DelivN | 2.00 (2.00, 3.00) | 2.00 (2.00, 3.00) | 2.00 (2.00, 3.00) | 3.00 (2.00, 4.00) | <0.001 |
| Systemic estrogen use | EstrSys | | | | | 0.052 |
| Never | | 1,926 (84%) | 406 (82%) | 1,046 (85%) | 474 (86%) | |
| Previously | | 101 (4.4%) | 33 (6.7%) | 44 (3.6%) | 24 (4.3%) | |
| Now | | 256 (11%) | 55 (11%) | 146 (12%) | 55 (9.9%) | |
| Birth control pill ever use | BCPilEv | 945 (41%) | 143 (29%) | 760 (61%) | 42 (7.6%) | <0.001 |
| Birth control pill current use | BCPilCu | 95 (4.2%) | 9 (1.8%) | 86 (7.0%) | 0 (0%) | <0.001 |
| Currently regular menstruation | MensRegCu | 937 (41%) | 139 (28%) | 748 (61%) | 50 (9.0%) | <0.001 |
| Education | Educ | | | | | <0.001 |
| Primary school | | 1,034 (45%) | 281 (57%) | 334 (27%) | 419 (76%) | |
| High school | | 637 (28%) | 128 (26%) | 419 (34%) | 90 (16%) | |
| University qualifying exam | | 159 (7.0%) | 20 (4.0%) | 126 (10%) | 13 (2.4%) | |
| University < 4 years | | 279 (12%) | 41 (8.3%) | 218 (18%) | 20 (3.6%) | |
| University 4 years or more | | 174 (7.6%) | 24 (4.9%) | 139 (11%) | 11 (2.0%) | |

[a]Median (IQR); n (%).

[b]Kruskal-Wallis rank sum test; Pearson's Chi-squared test; Fisher's exact test.

This table presents a selection of the available lifestyle-related variables (the variables showing the biggest differences across lifestyle-defined clusters) available for the study cohort and their distributions. The full table is presented in Supplementary Data 1.

Values are reported as median (IQR) or as number of individuals (%), both for the whole population (Overall), and for each cluster separately (Columns 4-6). P-value for testing if differences across clusters are statistically significant. All tests are two-sided.

## Metabolic variations explained by lifestyle-related factors

Ridge regression was applied to assess the amount of variance explained by the variation in lifestyle-related factors for each metabolite. The results are presented in Fig. 2, in which molecules with negative values for variance explained have been removed, while a complete overview is presented in Supplementary Data 2. Overall, total plasma Apo-B, cholesterol, triglycerides, and lipoproteins within the VLDL and LDL subfractions (except VLDL-5) had the highest explained variance, with a maximum of 30% for free cholesterol in IDLs, averaged over the ten cross-validation (CV) folds (Fig. 2A, Supplementary Data 2). Most VLDL and IDL- related parameters had variance explained exceeding 20%. In contrast, LDL and HDL-related parameters (except LDL-1), in addition to Apo A1 and Apo A2 in total plasma, had substantially lower amount of variance explained ( < 10% on average). For LDLs, the highest amount of variance explained was within the larger LDL subfractions LDL-1 and LDL-2 ( > 20% for triglycerides, 10–20% for remaining parameters), compared to the smaller subfractions LDL-3 to 6 (10–20% for triglycerides; 0–10% for remaining parameters).

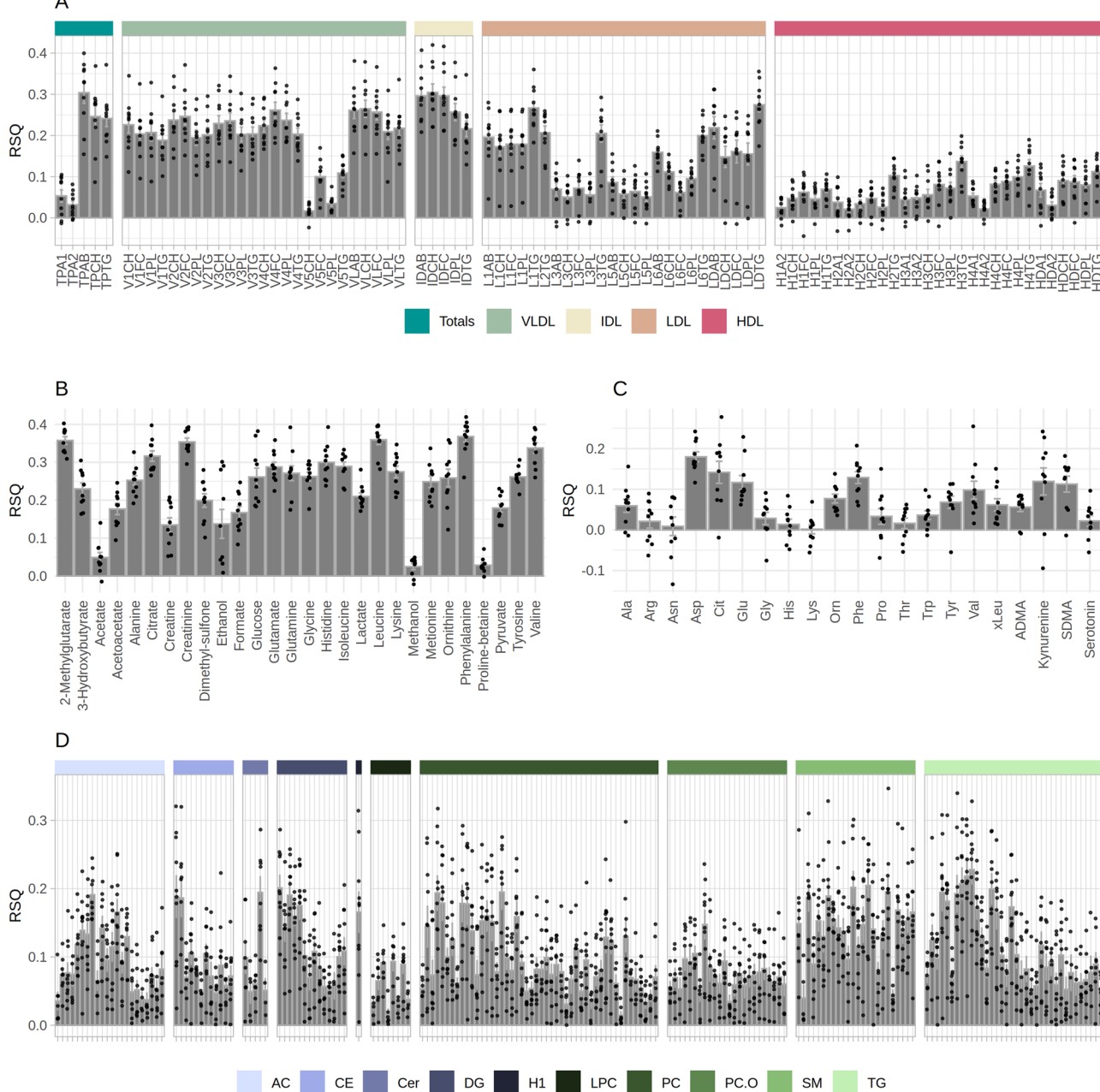

**Fig. 2 | Variance explained in individual metabolites explained by lifestyle-related variables.** Variance explained (RSQ) by lifestyle related factors in NMR-measured lipoprotein subfractions (**A**) and metabolites (**B**), and MS-measured metabolites (**C**) and lipids (**D**). MS-measured lipids (**D**) are ordered in the direction of an increased number of total carbons and number of double bonds, going from the left to the right. Molecules with a negative variance explained have been removed from the figures. TP: Total plasma, VLDL: Very-low-density lipoprotein, IDL: Intermediate-density lipoprotein, LDL: Low-density lipoprotein, HDL: High-density lipoprotein, CH: Cholesterol, FC: Free cholesterol, PL: Phospholipids, TG: Triglycerides, AB: Apolipoprotein-B, A1: Apolipoprotein-A1, A2: Apolipoprotein-A2; Ala:

Alanine; Arg: Arginine; Asn: Asparagine; Asp: Aspartate; Cit: Citrulline; Glu: Glutamate; Gly: Glycine; His: Histidine; Lys: Lysine; Orn: Ornithine; Phe: Phenylalanine; Pro: Proline; Thr: Threonine; Trp: Tryptophan; Tyr: Tyrosine; Val: Valine; xLeu: Leucine+Isoleucine; ADMA: Asymmetric dimethylarginine; SDMA: Symmetric dimethylarginine; AC: Acylcarnities; CE: Cholesteryl esters; Cer: Ceramides; DG: Diglycerides; H1: Hexoses; LPC: Lysophosphatidylcholines; PC: Phosphatidylcholines; SM: Sphingomyelins. Error bars show the mean and standard error of the estimated RSQ values over 10-fold cross-validation (n = 10 independent model training and testing sets).

Less variance was explained for small molecular metabolites ( < 20% for all but two metabolites), as shown in Fig. 2B, C. For NMR-measured metabolites, the highest amount of variance was explained for glutamate, leucine, isoleucine, phenylalaninie, and lysine, which are all amino acids. Among MS-measured metabolites, six had on average explained variance exceeding 10%, and only aspartate (Asp) had more than 15% explained variance. For MS-measured lipids, the highest variance explained was obtained for

sphingomyelins (SMs), followed by triglycerides (TGs) and diglycerides (DGs), many of which were in the range of 10–20%. In contrast, lysophosphatidylcholines (LPCs) and ether-linked phosphatidyl-cholines (PC-Os) had the lowest amount of variance explained (less than 10% on average). Interestingly, for ceramides, diglycerides, phosphocholines, and triglycerides, a trend was observed with a decrease in variance explained as the number of total carbon atoms in the fatty acid chains increased within each lipid class.

## Associations between circulating markers and lifestyle-related variables

Correlations between lifestyle-related variables and NMR-measured lipoproteins and metabolites were investigated through Spearman correlation analysis. Correlations ranged between -0.45 and 0.54, and the highest number of significant correlations were within the IDL and VLDL subfractions (Fig. 3A, Supplementary Data 3). In specific, age, blood pressure, and body weight showed positive correlations with these subfractions, and a non-linear dose-response relationship was observed between the lifestyle-related variables age, BMI, systolic blood pressure, waist to hip ratio (WHR), and the metabolic markers total plasma cholesterol (TPCH), total plasma triglycerides (TPTG), and LDL cholesterol (LDCH) (Fig. 4A-E). Specifically, levels of LDL cholesterol, total cholesterol, and total triglycerides showed a gradual increase with age, with peak levels at the age of 64 years, followed by a gradual decrease. For WHR, a peak concentration in total plasma triglycerides was observed at WHR of 1, with a gradual decrease for higher WHR values. Length of education, self-reported health, regular menstruation, and current use of birth control pills showed negative correlations to VLDL, IDL, and LDL subfractions. HDL subfractions (except for HDL triglycerides) had a different correlation pattern than most of the lipoprotein subfractions and metabolites, and weak negative correlations were observed between HDL subfractions and overweight.

For NMR-measured metabolites, most of the significant correlations to lifestyle-related variables were observed among amino acids, and similarly, as for the lipoproteins, their levels were significantly correlated with age, blood pressure, and body weight, and negatively correlated with education, self-reported health, regular menstruation, alcohol, and height. (Fig. 3B). Interestingly, a similar non-linear relationship as observed for the lipoprotein subfractions was observed between several amino acids (alanine, glutamate, leucine, and valine) and the abovementioned lifestyle-related variables (Fig. 4F–J). In specific, these amino acids had peak levels at WHR equal to 1, followed by a gradual decrease.

Weaker correlations were observed between lifestyle-related variables and MS-measured lipids and metabolites (Fig. 3C, D), however, the same lifestyle-related variables had positive or negative correlations with the MS-measured variables. Importantly, correlation patterns of aggregated levels of triglycerides (TG) and diglycerides (DG) closely resembled correlation patterns observed for the triglyceride-rich VLDL and IDL subfractions, as expected. For aggregated lipid concentrations, a non-linear dose-response relationships was often observed for age, BMI, WHR, and systolic blood pressure (Fig. 4K-T). Interestingly, acylcarnites, cholesteryl esters and sphingomyelins were quite constant, diglycerides and triglycerides increased, while lysophosphatidylcholines (LPC-Os) decreased with increased WHR and BMI. For MS-measured metabolites, a strong decrease in glycine with increased WHR was observed by visual inspection, while it was quite constant over different ranges of BMI (Fig. 4V, X).

## Cluster analysis

To determine metabolic profiles related to specific demography and lifestyles, the study population was clustered into three clusters based on available lifestyle-related variables. For clustering purposes, age and measurements of serum lipids/metabolites at participation in HUNT2 were left out in order to avoid clusters dominated by these factors. A proportion of the included lifestyle-related variables consisted of binary variables referring to self-reported health conditions and with only a few occurrences for each health condition. Therefore, to obtain more robust clusters, categorical variables were given less weight in the k-prototype clustering algorithm than numerical variables ($\lambda = 0.01$).

The three obtained clusters consisted of 494 (cluster 1), 1236 (cluster 2), and 553 (cluster 3) individuals (Fig. 5). Lifestyle-related variables, when projected onto a principal component scores plot, showed that the direction of the highest variation was defined by overweight, blood pressure, and reproductive variables (Figure S2), and even though age was excluded for clustering purposes, the first principal component had high negative correlation with age ($\rho = -0.70$). Cluster-specific differences in the HUNT2

variables are summarized in Table 1 and show that the main characteristic of cluster 1 is obesity, the main characteristic of cluster 3 is old age, while cluster 2 consists of the youngest and healthiest participants. Cluster 1 has a median BMI of 32 kg/m$^2$, median age equal to 54.6 years, and median systolic blood pressure of 142 mmHg, and consists mainly of individuals with low physical activity levels, low education, and good or not-so-good self-reported health. Cluster 2 has a median BMI of 24 kg/m$^2$, median age of 44.6 years, median systolic blood pressure of 122 mmHg, and has the highest number of participants who have higher education, are physically active, and/or report good self-reported health. Cluster 3 has a median BMI of 27 kg/m$^2$ median age equal to 68.3 years, a median systolic blood pressure of 157 mmHg, and consists mostly of individuals with only primary education. Compared to cluster 1, individuals in cluster 3, report slightly higher levels of physical activity, and better self-reported health .

## Cluster-specific metabolic signatures

Figure 6A (Supplementary Data 4) shows the average percentage differences in concentrations of lipoprotein subfractions in participants of lifestyle-defined clusters 1 and 3 compared to the youngest and most healthy cluster (cluster 2). Both increased body weight and increased age contribute to increased VLDL and IDL levels, while LDL and HDL levels were more similar across all three clusters, except for HDL triglycerides, which were lowest in cluster 2. Most of the VLDL levels were, on average, more than 50% higher in participants of cluster 1 and 3 compared to cluster 2. Furthermore, participants in cluster 1 had slightly higher concentrations of VLDL-1s and IDL triglycerides, while participants in cluster 3 had higher levels of LDLs. Figure 6E, I, and M show the distribution of the concentrations of selected lipoproteins in the three clusters and prove that the range of the lipoproteins is much wider for participants in clusters 1 and 3 compared to cluster 2 (broader peaks). Findings on MS-measured lipids showed that specifically diglycerides and triglycerides were elevated in clusters 1 and 3, compared to cluster 2, with a relative difference in concentration of over 30% (Fig. 6D). Several acylcarnies were highest in cluster 3, suggesting a characteristic of increased age, while ether-linked phosphatidylcholines (PC-Os) were decreased in cluster 1 only, which is therefore a potentially obesity-related characteristic. More specifically, PC-Os were negatively correlated with obesity (Fig. 3C), in contrast to the remaining lipids. In addition, levels of acylcarnitines, ceramides, lysophosphatidylcholines, sphingomyelins, cholesteryl esters, and phosphatidylcholines were highest in cluster 3. By investigating the distribution of selected lipids in the three clusters, as illustrated in Fig. 6 H, L, and P, we observed that these were more overlapping and similar in shape than for lipoproteins, except for triglycerides.

Several amino acids were upregulated in clusters 1 and 3 compared to cluster 2 (Fig. 6B, C), however, the relative differences in concentrations were lower than for lipoproteins. Only a few metabolites had average concentrations that were 15% or higher compared to cluster 2. For NMR-measured metabolites, acetoacetate, leucine, and isoleucine showed the largest inter-cluster variation, and had the highest levels in cluster 1 (Fig. 6 F, J, N). In contrast, concentrations of glycine, glutamine, proline-betaine, and histidine were quite constant across the clusters. For MS-measured metabolites, metabolites with the highest inter-cluster variation were glutamate (highest levels in cluster 1, Fig. 6G), methionine sulfoxide (Fig. 6K) and kynurenine (both elevated in clusters 1 and 3), and citrulline (highest levels in cluster 3). Interestingly, serotonin was decreased in clusters 1 and 3 compared to cluster 2 (Fig. 6O).

The most disticint differences between clusters 1 and 3 include: highest levels of VLDL-1 and IDTG, several amino-acids except for dimethyl-sulfone, and lowest levels of putrescine, seretonin, and ether-linked phosphatidylcholines (PC-Os) in cluster 1; and highest levels of LDLs, citrate, sarcosine, symmetric dimethylarginine (SDMA), trans-4-Hydroxyproline (t4.OH.Pro), acetylcarnities (Acs), ceramides (Cers), and most phosphatidylcholines (PCs) and Sphingomyelins (SMs) in cluster 3.

Since the participants in this study were initially selected for a breast cancer association study, data on breast cancer incidence within a 21 year follow-up period were available. We thus investigated breast cancer

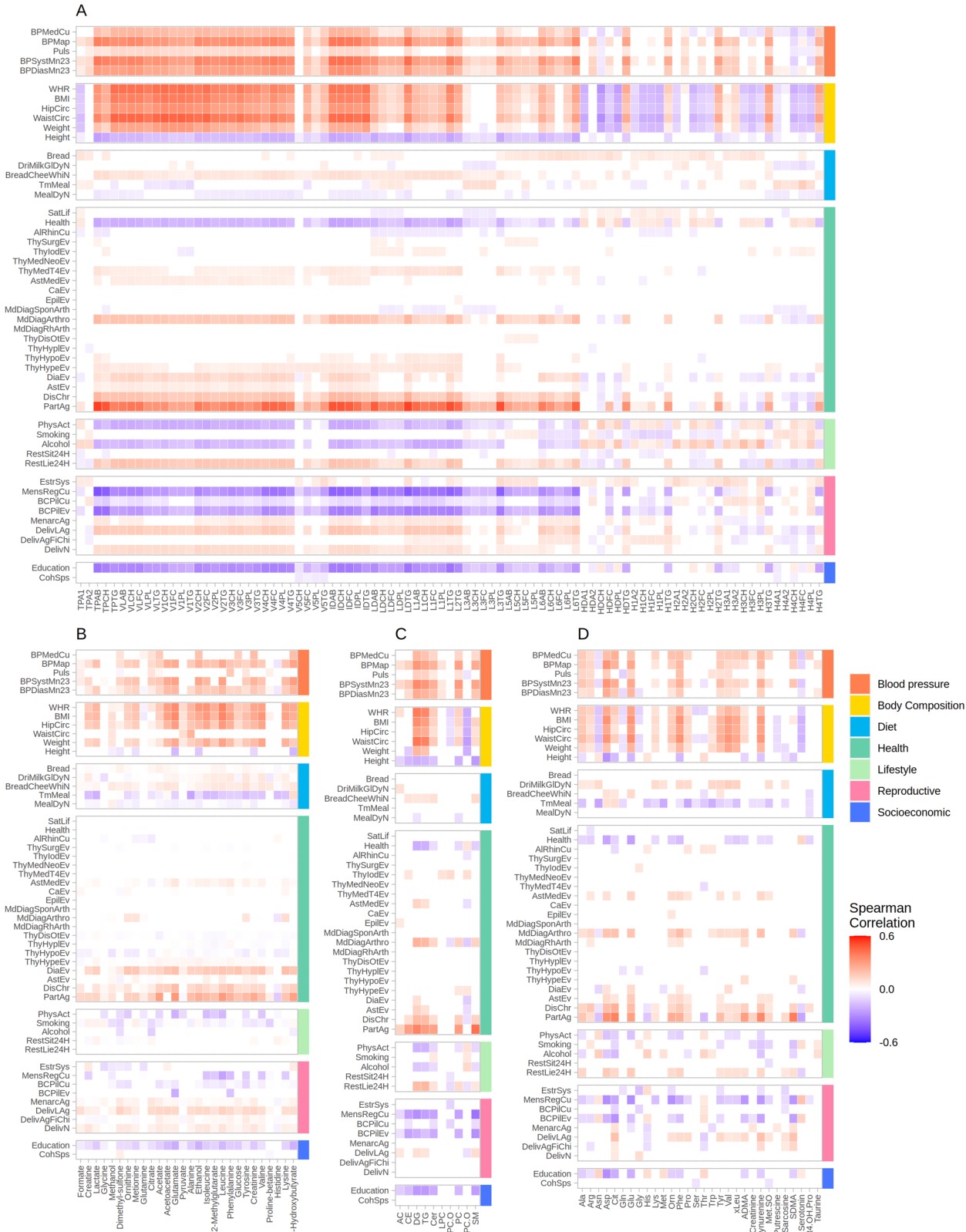

**Fig. 3 | Correlations between metabolites and lifestyle-related variables.** Spearman correlations between lifestyle-related variables (numerical or ordinal) and (**A**) NMR-measured lipoprotein subfractions; (**B**) NMR-measured metabolites; (**C**) Aggregated levels of MS-measured lipids and (**D**) MS-measured metabolites. TP: Total plasma, VLDL: Very-low-density lipoprotein, IDL: Intermediate-density lipoprotein, LDL: Low-density lipoprotein, HDL: High-density lipoprotein, CH: Cholesterol, FC: Free cholesterol, PL: Phospholipids, TG: Triglycerides, AB: Apolipoprotein-B, A1: Apolipoprotein-A1, A2: Apolipoprotein-A2; AC: Acylcarnitines;

CE: Cholesteryl esters; DG: Diglycerides; TG: Triglycerides; LPC: Lysophosphatidylcholines; Cer: Ceramides; SM: Sphingomyelins; PC: Phosphatidylcholines; Ala: alanine; Arg: arginine; Asp: aspertine; Cit: Citrulline; Gln: Glutamine; Glu: Glutamate; Gly: Glycine; His: Histidine; Lys: Lysine; Met: Methionine; Orn: Ornithine; Phe: Phenylalanine; Ser: Serine; Thr: Threonine; Trp: Tryptophan; Tyr: Tyrosine; Val: Valine; xLeu: Leucine+Isoleucine; ADMA: Asymmetric dimethyllarginine; Met.SO: Methionine sulfoxide; SDMA: Symmetric dimethylarginine; t4.OH.Pro: trans-4-Hydroxyproline; Insignificant correlations are colored in white.

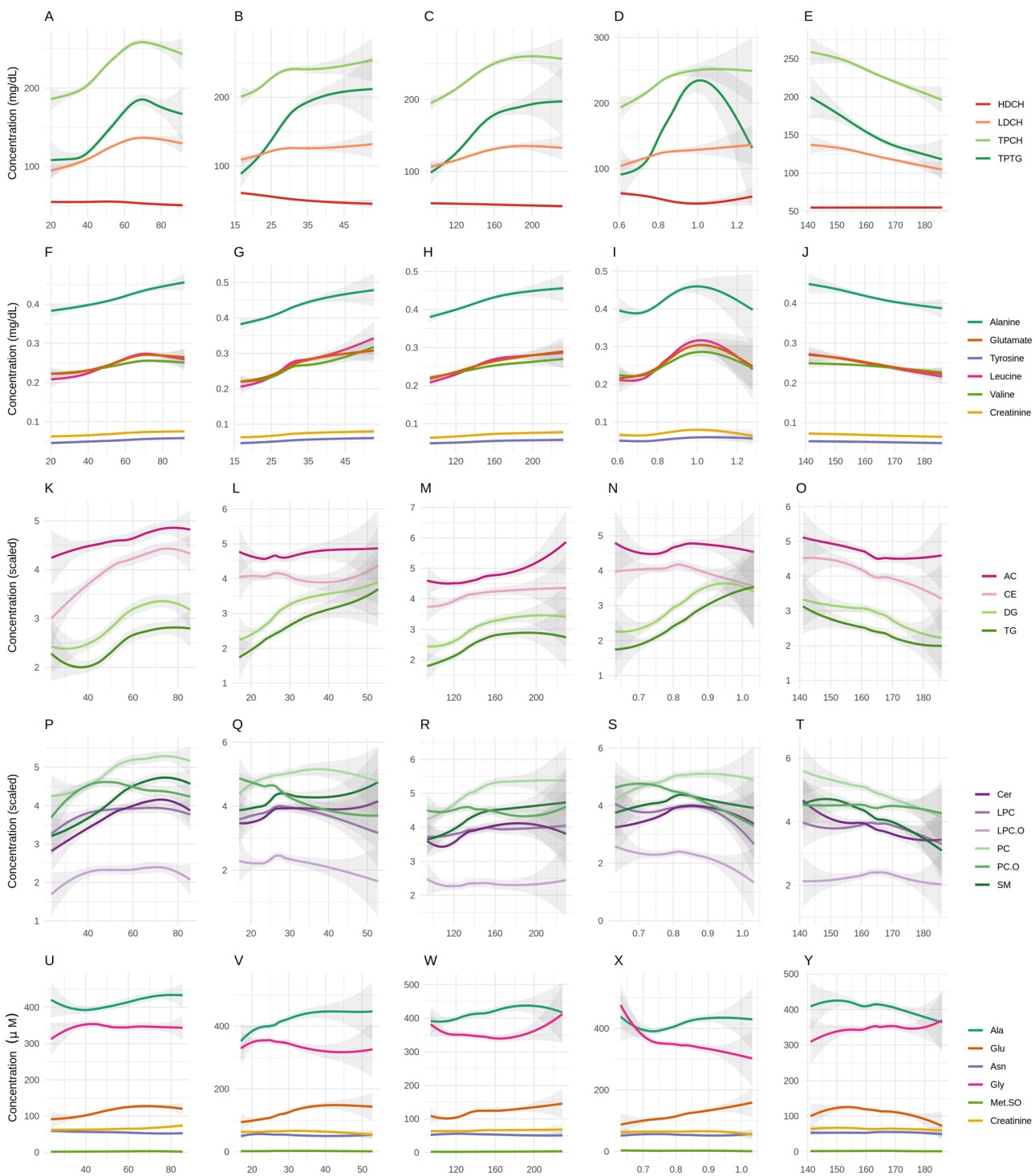

**Fig. 4 | Trends in metabolite levels in selected lifestyle-related variables.** Smoothed conditional means showing the relationship between selected lifestyle-related variables and a selection of NMR-measured lipoprotein subfractions (**A–E**); NMR-measured metabolites (**F–J**); Aggregated levels of MS-measured lipids and (**K–T**) and MS-measured metabolites (**U–Y**). BMI: Body mass index; WHR: Waist-to-hip ratio HDCH: High-density lipoprotein cholesterol; LDCH: Low-density lipoprotein cholesterol; TPCH: Total cholesterol; TPTG: Total triglycerides; AC: Acylcarnitines; CE: Cholesteryl esters; DG: Diglycerides; TG: Triglycerides; Cer: Ceramides; LPC: Lysophosphatidylcholines; PC: Phosphatidylcholines; SM: Sphingomyelins; Ala: Alanine; Glu: Glutamate; Asn: Asparagine; Gly: Glycine; Met.SO: Methionine sulfoxide.

incidence across the three clusters of participants and found that it was 48%, 49%, and 44%, in clusters 1, 2, and 3, respectively.

As an additional sensitivity analysis, we clustered the participants into three clusters based on all NMR-measured variables, using K-means clustering. The clusters are depicted in Figure S3, which shows that when projected onto a PCA scores plot, a gradual increase in an unfavorable serum-metabolic profile may be observed when going from NMR-defined cluster 3 to 1 (Figures S3A, B). We observe that the proportion of participants with the most unfavorable metabolic profile was the highest among participants within the lifestyle-defined clusters 1 and 3, while only 5% of

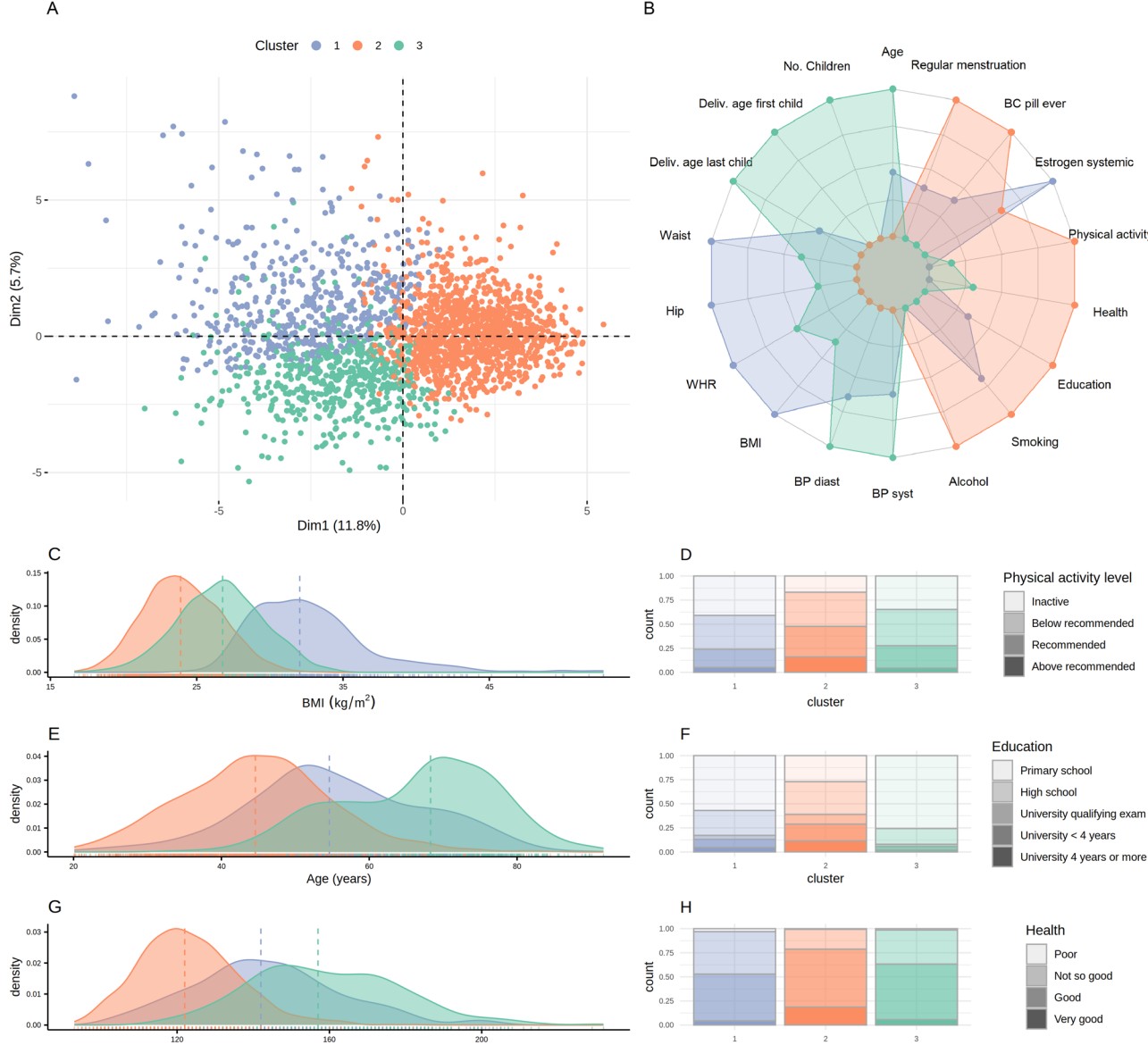

**Fig. 5 | Characteristics of participants in the lifestyle-defined clusters of the HUNT2 participants.** (**A**) The three clusters of participants projected onto a PCA scores plot. Both numerical and ordinal lifestyle-related variables were included in the PCA. (**B**) A visual summary of the main differences between the participants in the three clusters. The axis goes from the minimum (mean of the participants in the cluster with the lowest values) to the maximum (mean of the participants in the cluster with the highest values) for each lifestyle-related variable individually. The distribution of BMI (**C**), age at HUNT2 (**E**) and mean systolic blood pressure (**G**) in the three clusters is shown. Median values are depicted by the vertical dashed lines. Distributions of physical activity levels (**D**), education (**F**), and self-reported health (**H**) varied between the three clusters. PCA: Principal component analysis. BC: Birth-control; BP: Blood pressure; WHR: Waist-to-hip ratio; BMI: Body mass index.

participants within the lifestyle-defined cluster 2 had this metabolic profile (Figure S3-C).

## Discussion

Serum metabolomics provides a precise insight into the current health state of an individual, as it is affected both by preceding -omics levels (internal factors) and environmental/external factors. Its dynamic nature and sensitivity to sample handling, however, pose challenges for identifying robust disease biomarkers. Understanding which factors influence circulating metabolite levels may potentially lead to disease-preventive measures. We have previously shown that the serum metabolome is associated with breast cancer risk in an age-dependent manner[23,24], and in this study we investigated how other known breast cancer risk factors, as well as other lifestyle factors, influence the female serum metabolome. By utilizing two complementary analytical platforms, we investigated a wide panel of small-molecule metabolites, lipoproteins and lipids, and correlated them with lifestyle factors. Our analyses revealed that age is the largest single factor affecting the serum metabolic profile, and causes an unfavorable lipid profile, reflected in increased serum levels of most lipoproteins and lipids. However, age is not a modifiable factor. Interestingly, both increased body weight and systolic blood pressure, both potentially modifiable factors, contribute to an increase in the same markers.

In this study, we investigated how much of the total variance in single metabolites may be explained by breast-cancer related lifestyle variables, and found that around 20–30% variance could be explained within VLDL1-4 and IDL subfractions by these variables (Fig. 2A). For lipoproteins within the LDL subfractions, variance explained was lower, and negligible for HDL subfractions. Triglycerides were overall the lipids within the lipoproteins for which we obtained the highest variance explained. For MS-measured lipids (Fig. 2D), the variance explained was in general lower than for lipoproteins,

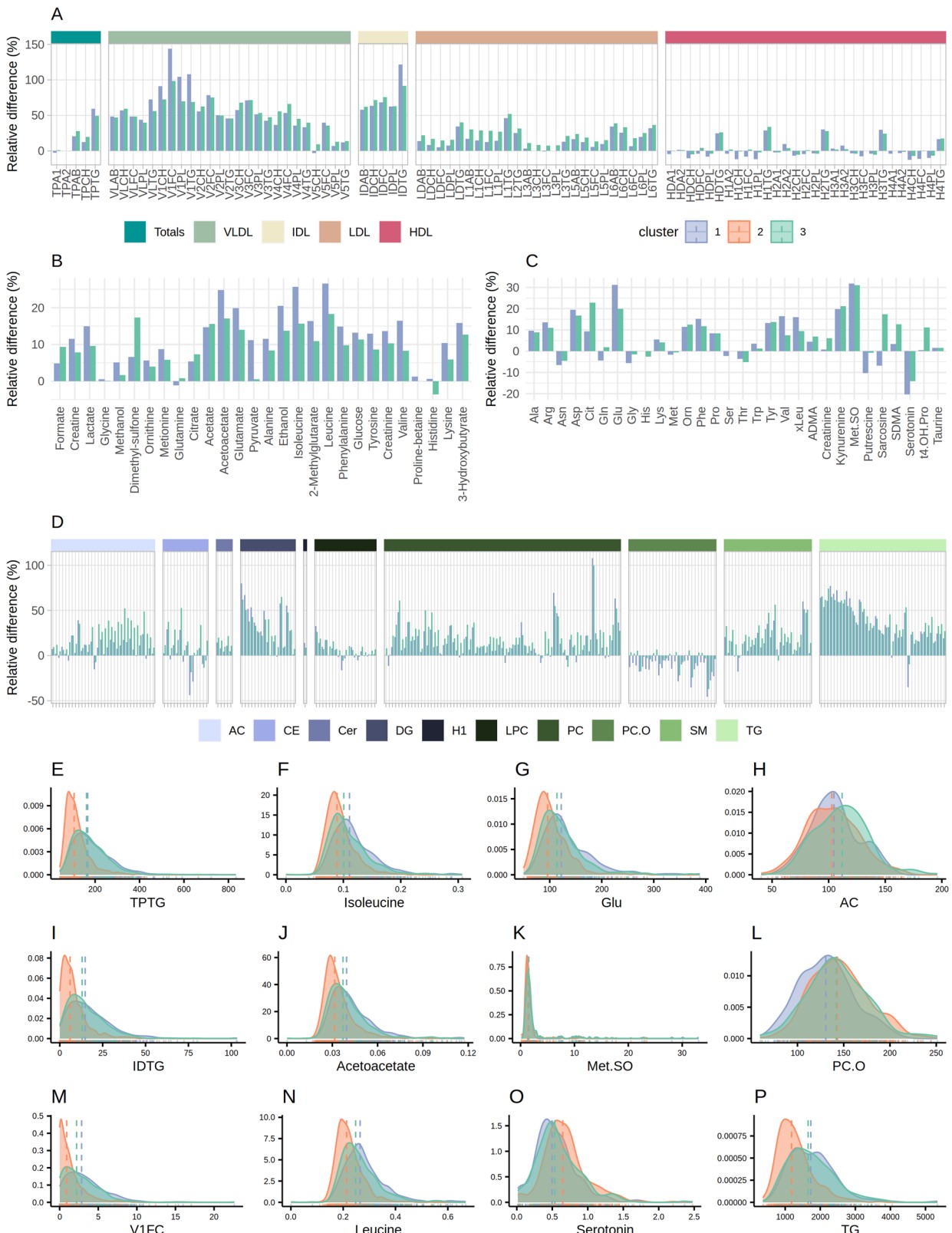

with values: less than 20% for all except 5 lipids. Sphyngomyelins were the lipids for which variance explained was the highest (range 15-20% for most parameters), while it was lowest for lysophosphatidylcholines. Only two metabolites (glutamate and leucine) had variance explained exceeding 20%, while 16 NMR-measured and 6 MS-measured metabolites had variance explained in the range of 10–20%, respectively (Fig. 2B, C). The generally

lower levels of variance explained for MS-measured parameters may partially be explained by the smaller size of the cohort with available MS data (815 individuals). Our findings are comparable to other studies that have investigated sources of variation in the circulating metabolome. For example, a study by Lau et al. showed that population-specific variance, combining information on age, sex, BMI, ethnicity, dietary information, and

**Fig. 6 | Metabolic differences among participants in the clusters of the HUNT2 participants.** Mean relative differences (%) in concentrations of NMR-measured serum lipoproteins (**A**) and metabolites (**B**) and MS-measured metabolites (**C**) and lipids (**D**) among participants of lifestyle-defined clusters 1 (blue) and 3 (green) compared to participants in cluster 2 (orange). Distributions of selected NMR-measured lipoprotein subfractions (**E, I, M**) and metabolites (**F, J, N**), and MS-measured metabolites (**G, K, O**) and aggregated lipids (**H, L, P**) in the three clusters of participants. Median values are depicted by the vertical dashed lines. TP: Total plasma, VLDL: Very-low-density lipoprotein, IDL: Intermediate-density lipoprotein, LDL: Low-density lipoprotein, HDL: High-density lipoprotein, CH: Cholesterol, FC: Free cholesterol, PL: Phospholipids, TG: Triglycerides, AB: Apolipoprotein-B, A1: Apolipoprotein-A1, A2: Apolipoprotein-A2; Ala: Alanine; Arg: Arginine; Asp: Aspartate; Cit: Citrulline; Gln: Glutamine; Glu: Glutamate; Gly: Glycine; His: Histidine; Lys: Lysine; Met: Methionine; Orn: Ornithine; Phe: Phenylalanine; Ser: Serine; Thr: Threonine; Trp: Tryptophan; Tyr: Tyrosine; Val: Valine; xLeu: Leucine+Isoleucine; ADMA: Asymmetric dimethylarginine; Met.SO: Methionine sulfoxide; SDMA: Symmetric dimethylarginine; t4.OH.Pro: trans-4-Hydroxyproline; AC: Acylcarnitines; CE: Cholesteryl esters; Cer: Ceramides; DG: Diglycerides; H1: Hexoses; LPC: Lysophosphatidylcholines; PC: Phosphatidylcholines; SM: Sphingomyelins.

country of origin, explained a median of 9% variance in serum metabolites in a cohort of European children[30]. Other studies have investigated the influence of the microbiome, diet, lifestyle, clinical data, and genetics on the variation in the human serum and plasma metabolome[5,31], and found that the strongest factor affecting the serum metabolome was the diet, followed by clinical data, microbiome, genetics, time of day, and finally lifestyle[6]. Combined, they obtained a median variance explained exceeding 40%. The strongest factor affecting the plasma metabolome was the microbiome, followed by diet, a combination of age, sex, BMI and smoking, and genetics[31]. When combined, these factors explained 25% variance. In our study, we lacked detailed dietary information, as well as genetic data and data on the microbiome composition, thus the influence of these factors could not be investigated. Our findings indicate that IDL and VLDL levels reflect the overall lifestyle and that their levels may be manipulated by a weight-reduction or decrease in blood pressure. However, the causality of these findings should be confirmed using causal inference methods, such as Mendelian randomization[32,33].

Our results showed that several metabolites are weakly correlated with age, obesity, and systolic blood pressure, whereas specific lipoprotein subfractions within VLDL and IDL have stronger correlations with these factors (Fig. 3A–D). Furthermore, we observed that the metabolic profile is influenced by the presence of regular menstruation, use of birth control pills, length of education, and self-reported health, with decreased levels of most lipoproteins and metabolites with increased levels of the abovementioned factors. Associations between lifestyle-related variables to lipoprotein subfractions and MS-measured lipids were in agreement, reflected in similar correlation patterns for the triglyceride-rich lipoproteins VLDLs and IDLs, as seen for MS-measured triglycerides (Fig. 3A, C). Noticeably, no strong correlations were observed for lysophosphatidylcholines (LPC-Os), which levels have been associated with a lower risk of several cancers, including breast cancer[34], indicating potentially robust cancer predictive markers (Fig. 3C).

Obesity-induced changes in lipid metabolism are in general unfavorable, and our study confirms previously reported non-linear relationships between adiposity and lipoproteins[25,35–37]. We found that in specific VLDL1-4s and IDLs were correlated with increased adiposity, while HDLs (except for triglycerides) showed weak negative correlations (Fig. 3A). VLDL-5s (except for V5FC) were the only lipoproteins within the VLDLs that had only negligible correlations with lifestyle-related factors. Total plasma Apo-B, which is found in lipoproteins originating from the liver (LDLs, VLDLs, and IDLs, but not HDLs) was strongly correlated with age, adiposity, and blood pressure. This protein is associated with plaques that cause atherosclerosis[38], and has been suggested as a better marker for coronary heart disease than LDL cholesterol[39]. Correlations were in general higher for waist circumference than both BMI, WHR, and hip circumference, indicating that increased VLDL1-4, IDL, and Apo-B levels are specifically associated with increased abdominal fat. For aggregated MS-measured lipids we observed positive correlations between di- and triglycerides and increased body fat (Fig. 3C), while esterified phosphatidylcholines (PO-Os), were negatively correlated to body fat. Interestingly, sphingomyelins were negatively correlated with height, while very weak correlations were observed with remaining body measurements. We also confirmed positive correlations between adiposity and several circulating metabolites[25,37], in specific amino acids, glucose, and lactate (Fig. 3A, D). Elevated levels of

branched-chain amino acids have been linked with an increased risk of diabetes, and a synergistic relationship between obesity-related insulin resistance, diabetes, and cancer has been proposed[40].

High blood pressure and cholesterol are linked; arteriosclerosis occurs when cholesterol plaque and calcium accumulate, arteries stiffen and narrow, placing increased strain on the heart to pump blood. Consequently, blood pressure rises to unhealthy levels. Here we provide insight into the relationship between blood pressure and the serum metabolic profile. Specifically, we demonstrate that lipoprotein subfraction levels increase with increased systolic blood pressure, where the steepest increase was in the region 110–160 mmHg for total plasma triglycerides, and total and LDL cholesterol (Figs. 4C and 4M). Our findings align with previously reported findings of an association between elevated cholesterol levels (except HDL cholesterol) and hypertension[41–43]. We show that also metabolites are correlated with blood pressure, however, the correlations were much weaker (Fig. 4H, W). Systolic blood pressure is highly correlated with increased age, thus it is difficult to separate the influence of these two factors on the metabolic profiles.

Menopause causes hormonal changes with decrease in estradiol which could cause estrogen deficiency[44]. This leads to dysregulated lipid metabolism[45], and we have previously shown that several lipoprotein levels (especially within the VLDL, IDL and LDL subfractions) are increased in postmenopausal compared to pre-menopausal women[24]. In the present study we observe lower levels of lipoproteins among participants who reported current regular menstruation (Fig. 3A). We also observed lower lipid levels among women using birth control pills (ever use), but no or very weak effect of current use of birth control pills was observed. Other studies have in general reported higher lipid levels in oral contraceptive users[46–48]. We, therefore, believe that the observed lowered lipid values can be explained by age differences, as widespread use of birth control pills was still in its infancy at the time the eldest HUNT2 participants were premenopausal.

Other factors that were found to influence the serum metabolic profile were length of education, self-reported health, time since the last meal (for metabolites only), and physical activity levels (Fig. 3A–D). These factors were negatively correlated with the same lipoprotein subfractions and lipids that were positively associated with increased age and body weight and may thus be included as a strategy for maintaining a healthy lipid profile. Our findings align with previously reported findings on the metabolic profiles of socio-economic position, where low education attainment was associated with higher levels of VLDLs and triglycerides in HDLs, and lower levels of HDLs in basic-adjustment analysis[49]. The authors show that many of these associations were mediated through risk factors or dietary indicators, and in risk-factor adjusted analyses, higher levels of triglycerides in small HDLs and lower levels of apo-A1 and large-very large HDL particles (including levels of their respective lipid constituents) and cholesterol measures across different density lipoproteins were associated with low educational attainment.

As many of the lifstyle factors investigated in this study are correlated, it is difficult to separate the influence of age and obesity on the metabolic profile from influence of other, correlated lifestyle-related factors. Because of this, the study participants were clustered into three groups based on lifestyle-related factors, and we report distinct metabolic signatures for each cluster. Participants in cluster 1 (whose main characteristic is obesity) and

cluster 3 (characterized by old age) had much more similar metabolic profiles when compared to participants in cluster 2 (youngest and normal-weight) (Fig. 6). The intriguing aspect lies in the metabolic health similarity observed between clusters 1 and 3, despite their diverse lifestyle variables. Some differences are worth noticing: cluster 1 had the highest VLDL-1, IDL triglyceride, branched-chain amino acid, and PC-O levels, suggesting these may be obesity-specific markers (Fig. 6). Cluster 3 had the highest levels of dimethyl-sulfone, citrate, sarcosine, and acylcarnitines (AC) levels, suggesting these are markers specific to aging. Both clusters 1 and 3 had lipid profiles which align with increased risk of developing cardiovascular diseases, thus we hypothesize that obesity accelerates metabolic aging.

The low incidence of breast cancer within cluster 3 compared to cluster 2 might be surprising, considering known breast cancer risk factors. However, this difference is not statistically significant (P-val = 0.27). It is important to note that the time-to-diagnosis was shorter for participants in clusters 1 and 3 than cluster 2, and that the lifestyle variables were collected up to 18 years prior to breast cancer incidence. In other words, this suggests that reaching a high age while preventing severe overweight, such as participants in cluster 3, reduces breast cancer risk. These participants have also reported the highest number of pregnancies and scored low on alcohol consumption and smoking, factors, which are associated with lower breast cancer risk.

A major strength of this study is the extensive metabolic profiling, including both NMR-measured metabolites and lipoproteins, and MS-measured metabolites and lipids, giving a high coverage of molecules. Nevertheless, due to a previously described contamination in the serum samples that interfered with one of the lipid peaks in the NMR spectrum[24], most parameters from LDL-2 (except triglycerides) and all parameters from LDL-4 subfractions could not be investigated in this study. Another strength is the size of the study cohort and the high number of factors investigated including demography, lifestyle and socio-economic factors, and anthropometric and clinical measurements. By focusing on machine learning methods that can handle mixed data types, we optimized utilization of the data. Together with the large metabolite coverage, measured by two complementary analytical platforms, this allowed us to investigate how different factors are correlated with the serum metabolome in detail. The results of this study have implications for breast cancer and other female cancer types with shared risk factors, and other medical conditions linked to disrupted metabolism, such as metabolic syndrome, diabetes, or cardiovascular health.

Some limitations need to be addressed. Participants in this study were initially selected for a breast cancer case-control study to investigate associations between circulating metabolites and lipoproteins and long-term breast cancer risk, and as an effect, the study cohort may not be completely representative of an average female population. The serum samples were collected non-fasting, and dietary information was very limited for the individuals, thus the effect of dietary intake on the levels of the measured metabolites, lipoprotein subfractions, and lipids could not be investigated. Similarly, self-reported data on medication usage was available for these participants, but with limited quality. Moreover, the samples were collected over a period of two years and differences in sample handling procedures, such as differences in time from sample collection to centrifugation cannot be ruled out. It has previously been shown that a delay in centrifugation alters the levels of some metabolites[16], and that storage of plasma for up to 7 years has negligible impact on the metabolic profile[50], while long-time storage may affect the levels of some lipids, amino-acids, and hexoses[51]. Importantly, these effects are randomly distributed over the samples and are commonly present for biobank samples.

In summary, we have in this study investigated sources of heterogeneity in the female serum metabolome, in light of both breast-cancer-related and other lifestyle variables. We have shown that up to 30% of the variation in individual serum metabolites of females could be explained by these factors, and that age, increased body size, and high blood pressure all contribute to a more unfavorable lipid profile. The strongest findings were observed for triglyceride rich VLDL and IDL subfractions, which increased both with increased age and with adiposity. Observed patterns for MS-measured triglycerides on a subcohort of the full cohort supported these findings. Interestingly, several metabolites showed similar trends in concentrations as VLDLs. In contrast, HDL subfractions were weakly correlated with lifestyle-related variables, and the variance explained for HDLs was low. Taken together, these findings suggest that VLDLs and IDLs are more reflective of lifestyle and current health status. In addition, obese participants have a metabolic profile similar to elder individuals, indicating that obesity may accelerate metabolic aging.

## Methods

### Study population

Participants in this study were initially selected for a breast cancer case-control study to investigate associations between circulating metabolites and lipoproteins and long-term breast cancer risk[24], based on the second wave of data gathering in the Trøndelag Health study (the HUNT study)[52,53]. The HUNT study is a prospective, longitudinal, population-based health study, conducted in Trøndelag, Mid-Norway since 1984, with repeated measures every tenth year: HUNT1 (1984-86), HUNT2 (1995-97), HUNT3 (2006-08) and HUNT4 (2017-19). It includes self-reported questionnaire-based health data, clinical examinations, and biological sampling from over 100,000 individuals.

In the study by Debik et al. all HUNT2 female participants with a later breast cancer diagnosis within a 22-year follow-up period were identified (n = 1208), by matching HUNT2 individuals with the Norwegian Cancer Registry in 2019. An age-matched control that remained breast cancer-free during the follow-up period was randomly selected for each case. Serum samples from these women, collected between 1995-97, were stored at -80 °C until analysis. In years 2019–2022, the samples underwent comprehensive metabolic profiling; nuclear magnetic resonance (NMR) spectroscopy was performed on all samples, and quantitative high-resolution mass spectrometry (MS) analysis using the Absolute IDQ p400 HR kit (Biocrates Life Sciences AG, Innsbruck, Austria) was later performed on a selected sub-cohort, n = 906[23]. This subcohort was selected so that that the number of case-control pairs increased proportionally with the time to diagnosis in the range 0–15 years[23]. Details on the metabolic profiling can be found elsewhere[23,24]. All HUNT participants have completed a written informed consent form, and the study was approved by the Ethics Committee of Central Norway (REK numbers #1995/8395 and #2017/2231). All ethical regulations relevant to human research participants were followed. Participants with diagnosed breast cancer within 3 years from sample collection were excluded from this study, leaving n = 2283 women with available NMR metabolomics data, and n = 815 women with available combined NMR and MS metabolomics data. All included participants were considered healthy.

**NMR measurements.** In short, lipoprotein subfractions were quantified using the fully automated Bruker IVDr Lipoprotein Subclass Analysis (B.I.LISA™) panel, from Bruker BioSpin. The Bruker IVDr Lipoprotein Subclass Analysis (B.I.LISA™) panel was used to quantify lipoprotein subfractions. It reports lipid concentrations in total serum and in four main lipoprotein classes (VLDL, IDL, LDL, HDL) and 15 subclasses. It also provides serum levels of apolipoproteins (Apo-A1, Apo-A2, Apo-B), 12 calculated parameters (ratios of LDL-CH/HDL-CH and Apo-B/Apo-A1), and particle numbers of total serum, VLDL, IDL, LDL and LDL 1 − 6, totaling 112 lipoprotein subfractions. However, due to contamination in the serum samples, some subfractions, mostly from LDL-2 and LDL-4 particles, were excluded from further analysis, along with calculated parameters and particle numbers, leaving 89 lipoprotein subfractions. Small-molecular metabolites were quantified by integrating respective areas under metabolite peaks in CMPG spectra, after careful spectra preprocessing. The peaks were adjusted for T2 relaxation times. The NMR experiments were performed at two different labs, and although the same protocol was followed, a slight batch effect in the lipid profiles was observed. Since this effect was negligible and non-systematic, it was not accounted for in this study. Importantly, samples were run in a

completely randomized order across the two labs. The proportion of variance introduced by the NMR measurements has been assessed through coefficients of variation (CVs) based on 46 quality control samples prepared from pooled serum samples of healthy controls, as previously described[24]. CVs of the metabolites were below 15% and below 20% for 23 and 26 of the metabolites, respectively, and below 15% and 20% for 65 and 85, respectively, of the lipoprotein subfraction measurements.

**MS data.** Quantitative analysis of more than 400 metabolites was performed using the Absolute IDQ p400 HR kit (96-well plates; Biocrates Life Sciences AG, Innsbruck, Austria). The kit combined direct flow injection for eight groups of lipids (phospholipids, glycerides, cholesterol esters or ceramides) and liquid chromatography (LC) high-resolution mass spectrometry (HRMS) for amino acids, biogenic amines, and a sum of hexoses.

Sample preparation for the MS measurements began with random placement of samples on measurement plates[23]. Subsequently, according to the protocol provided by the manufacturer, metabolites were extracted from 10 ml of blood serum (after applying the sample to a plate containing isotope-labeled internal standards and performing the on-plate chemical derivatization). Depending on the compounds analyzed, half of the extract was diluted in dedicated solutions. The metabolites were then analyzed using a system consisting of an Orbitrap Q Exactive Plus spectrometer (Thermo Fisher Scientific, Waltham, MA, USA) and a 1290 Infinity UHPLC liquid chromatograph (Agilent Santa Clara, USA), controlled by Xcalibur 4.1 software.

The obtained spectra and chromatograms were processed using Xcalibur 4.1 and MetIDQ DB110-2976 software (Biocrates Life Sciences, Innsbruck, Austria) dedicated by the kit manufacturer, obtaining concentration values in μM of individual metabolites present in specific samples.

After the array of metabolites concentrations was obtained, data analysis began with the detection and imputation of missing values. Metabolites for which measurements were missing completely at random (MCAR) in more or equal to 10% of samples were excluded from the dataset. Metabolites for which measurements were missed not at random (MNAR, i.e., values below the detection limit) in more or equal to 50% of samples were excluded from analysis. A threshold of 50% was adopted for MNAR values according to the recommendations of Chen and coworkers[54]. The remaining compounds ($n = 284$) were logarithmically transformed and then the batch effect was corrected using an empirical Bayes method, assuming that samples measured using a single 96-well sample preparation plate represent one batch[55]. In the next step, the MNAR values were imputed with random numbers from the normal distribution truncated to the range from 0 to the median of the LOD values determined for all measurement plates. In turn, MCAR values were imputed using the k-nearest neighbors method (the neighborhood ranking was determined within each group of samples separately, and the missing values were imputed with the average value of the level of a given metabolite for the 3 most similar samples). After removing metabolites with more than 50% missing values, the CVs of inter- and intra-batch QC measurements was calculated according to a method given by Zhang et al.[56]. Overall, the CV for 280 metabolites used in the quantitative analyses was below 15%. The median intra-batch coefficient of variation calculated for all metabolites in all QC samples ranged from 8 to 16%, and the median inter-batch CV was 18.73%.

**Variables related to demography and lifestyle**
Variables related to breast cancer risk were available for these participants, including variables on clinical parameters, demography, lifestyle, socio-economic factors, diet, and anthropometric measurements from the HUNT databank (collectively called lifestyle-related variables for simplicity). Variables related to physical activity levels were re-coded into previously published physical activity index to group individuals into four levels of physical activity: inactive, below recommended, recommended, or above

recommended, according to physical activity recommendations for adults[57–59]. Pulse and blood pressure measurements were measured 3 times and averaged for each individual. Body mass index (BMI) and waist-to-hip ratio (WHR) were calculated from the anthropometric measurements collected in HUNT2. After data cleansing and eliminating variables with more than 30% missing values 60 variables were retained for statistical analyses (Figure S1). A conservative threshold of 30% missingness was selected based on the characteristics of the available lifestyle variables in this study to ensure a good balance between robustness and retaining as many variables as possible for data analysis. Variables with high missingness were predominantly related to diseases and medication usage, which we considered to be 'missing not at random'. In contrast, variables with low missingness generally aligned with the assumption of being 'missing at random'.

**Statistics and Reproducibility**
Supplementary Fig. S1 shows the proportion of missing values for each of the lifestyle-related variables under investigation. For variables with less than 30% missing values, missing values were imputed using the random forest imputation algorithm implemented in the missForest[60] package in R. This method allows simultaneous imputation of missing values in mixed data types: for each non-complete variable, a random forest is created on observed data using all remaining variables and is used to predict missing values. This process of training and predicting is repeated in an iterative process, until a stopping criteria is met. The number of trees in each forest was set to 300 to ensure convergence of the out-of-bag (OOB) error rate. Principal component analysis (PCA) was applied to explore the directions of the largest variation in the serum metabolic profiles and to detect outliers. Spearman correlations were calculated between measured metabolites and numeric or ordinal lifestyle variables. K-prototype clustering[61] was employed to cluster the study participants into three distinct clusters based on lifestyle-related variables; serum metabolomics data were not included in the clustering. This clustering is suitable for mixed data types, as it is a weighted sum between the squared Euclidean distance (k-means clustering) and the total number of mismatches (k-mode clustering). Given $p$ variables, of which $m$ are continuous, the distance measure for k-prototype clustering is given by: $d(X, Y) = \sum_{i=1}^{m}(x_i - y_i)^2 + \gamma \sum_{i=m+1}^{p} \delta(x_i, y_i)$, where $\gamma$ is the weight that determines the trade-off between Euclidean distance of numeric variables and simple matching coefficient between categorical variables, and $\delta$ is equal to 0 if $x_i = y_i$, otherwise 1. Clustering was performed using the clustMixType package, and $\gamma$ was set to 0.01 based on a grid search over different $\gamma$ values (ranging from 0.01 to 5), where the silhouette score was used to evaluate the quality of clusters[62]. Numerical variables were auto-scaled prior to clustering. Age at participation in HUNT2, which is a major confounding factor for the serum metabolome, and serum measurements (cholesterol, triglycerides, non-fasting glucose, HDL cholesterol, and creatinine) performed at the biobank were excluded for the cluster analysis. These age and serum measurements were excluded as our aim was to define clusters based on lifestyle only, and then investigate cluster-specific differences in the serum profiles. Lifestyle-related variables and levels of measured metabolites were compared across the three clusters of participants, and significant differences between the clusters were tested by the Kruskal-Wallis rank sum test; Pearson's Chi-squared test, or Fisher's exact test. Nonparametric tests were chosen as the lifestyle-variables were not normally distributed, and all tests were two-sided. For metabolites, the Benjamini-Hochberg procedure was applied to adjust p-values for multiple tests. Ridge regression[63] models were used to assess the variance in each metabolite explained by the lifestyle variables using the glmnet package in R[64]. Both metabolite levels and numeric lifestyle-related variables were autoscaled before regression analysis. The Ridge regression models were implemented in a nested cross-validation loop. The optimal value for the penalty parameter λ was found through cross-validation on the training set, by a grid search with 100 evenly spaced λ values between 0 and 1. For metabolites with the highest amount of variance explained, lifestyle-related variables with the highest importance in the model were identified. Variance

explained ($R^2$) was calculated for each metabolite as 1 minus the residual sum of squares (variation in data not explained by model) over the total sum of squares (total variation in the data), i.e. $R^2 = 1 - SS_{res}/SS_{tot}$. All statistical analyses were performed in R version 4.3.1 (2023-06-16).

## Reporting summary

Further information on research design is available in the Nature Portfolio Reporting Summary linked to this article.

## Data availability

The Trøndelag Health Study (HUNT) has invited persons aged 13–100 years to four surveys between 1984 and 2019. Comprehensive data from more than 140,000 persons having participated at least once and biological material from 78,000 persons are collected. The data are stored in HUNT databank and biological material in HUNT biobank. HUNT Research Centre has permission from the Norwegian Data Inspectorate to store and handle these data. The key identification in the database is the personal identification number given to all Norwegians at birth or immigration, whilst de-identified data are sent to researchers upon approval of a research protocol by the Regional Ethical Committee and HUNT Research Centre. To protect participants' privacy, HUNT Research Centre aims to limit the storage of data outside HUNT databank, and cannot deposit data in open repositories. HUNT databank has precise information on all data exported to different projects and are able to reproduce these on request. There are no restrictions regarding data export given approval of applications to HUNT Research Centre. For more information, see http://www.ntnu.edu/hunt/data. Source data underlying Figs. 2, 5, and 6 are provided in Supplementary Data 2, 3, and 4, respectively. Source data underlying Figs. 3, and 4, and Supplementary Figs. S1-S3, consist of sensitive information on an individual level and therefore cannot be published.

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

## Acknowledgements

The Trøndelag Health Study (HUNT) is a collaboration between HUNT Research Centre (Faculty of Medicine and Health Sciences, Norwegian University of Science and Technology NTNU), Trøndelag County Council, Central Norway Regional Health Authority, and the Norwegian Institute of Public Health. This work has been supported by the Norwegian Financial Mechanism (2014-2021, JD, TFB, Project 2019/34/H/NZ7/00503).

## Author contributions

T.F.B. and P.W. obtained funding for the study. G.F.G, T.F.B., and P.W. designed the study and supervised the work. K.M., K.J., A.K., and J.D. acquired data. J.D. Analyzed the data and drafted the manuscript. All authors have reviewed and approved the final manuscript.

## Funding

 Olavs Hospital - Trondheim University Hospital).

## Competing interests

The authors declare no competing interests.
