## [Transparent Peer Review file · Communications Biology]

Sources of variation in the serum metabolome of female participants of the HUNT2 study

Corresponding Author: Dr Julia Debik

Version 0:

Reviewer comments:

Reviewer #1

(Remarks to the Author)

1. Sample Preprocessing and Batch Effects:

This work involves a considerable number of biological samples, potentially introducing batch effects in both NMR and MS spectra data. The "Method" section in the manuscript provides limited information regarding sample preprocessing. We suggest the authors to supplement this section with more detailed information on sample preprocessing steps, particularly focusing on how batch effects were mitigated. Authors should provide specific procedures for sample handling, including but not limited to sample collection, storage, preparation, and analysis, along with detailing how batch effects were addressed, such as employing quality control measures or correction methods.

2. Impact of Sample Storage Time:

The long timespan of sample collection in this study raises concerns regarding the impact of sample storage time, which is a critical factor influencing metabolite concentrations. Did the authors consider the effect of sample storage time during data preprocessing? If so, we recommend authors to elaborate on the methods used to exclude or correct for this factor in the Methods section. If not considered, authors should discuss this limitation in the Discussion section and explore its potential implications on the results.

3. Features and Quantitative Information in NMR Data:

Figure 1 indicates the presence of 112+28 features in the NMR data. Please provide more information about these 112 features, including the compounds or metabolites they represent, and describe how their quantitative information was obtained. Additionally, do the 28 features represent small molecule metabolites? If so, please explain the identification and quantification methods for these metabolites.

4. Consistency between NMR and MS Measured Metabolites:

The statement in the third paragraph of the Discussion section, "Findings between NMR- and MS-measured metabolites were consistent, confirming the validity of the molecular measurements," should be supported with more detailed comparative results. We suggest the authors to list the metabolites measured by both NMR and MS techniques, comparing their concentrations or expression levels, and discussing their correlation or consistency across the two methods.

5. Repetitive Measurement Experiments and Variations Introduced by Measurement Process:

Variance in the data may include variations introduced by the measurement process. To assess this impact, we recommend authors to conduct some repetitive measurement experiments and provide an analysis of the obtained data in the Results section. Authors can estimate the proportion of variance introduced by the measurement process and discuss its implications for data interpretation and result reliability.

Reviewer #2

(Remarks to the Author)

Debik et al present an overview of serum metabolic profiles of 2,283 females in the HUNT study in association with lifestyle factors.

The manuscript is overall, well written and within scope of the journal. The manuscript builds on existing literature to

understand whether metabolomics can provide insight into risk factors for breast cancer. While not novel per se, the study does have several strengths and provides confirmatory as well as novel insights on specific metabolites.

Although, as mentioned above, the manuscript was well written, the abstract was difficult to follow. For example, the conclusion speaks to cluster analysis comparing obese and older individuals, but this was not outlined as an objective or in the results. Indeed, the aims are not presented, nor is the relevance to breast cancer articulated.

Although I acknowledge that lifestyle is a term widely used in public/population health (I also used to use it), I would encourage the authors to reflect on the use of the term, and perhaps provide alternative wording. Suggested reading: <https://ukpublichealthnetwork.org.uk/lifestyle-a-plea-to-abandon-the-use-of-this-word-in-public-health/#:~:text='Lifestyle' is a loaded term,responsible for their own health.>

Additional comments by section are outlined below.

Methods

- The timing of lifestyle variable collection (exposure) is unclear and possibly problematic.
 - o 1984 start of cohort
 - “repeat measures every 10 years” (455)
 - but also “4 surveys collected between 1984 and 2019” (529)?
 - o 22 year follow up time
 - 1984 start would bring us to 2006
 - But if individuals were recruited over a 13 year period, then could get to 2019 end collection time.
 - o 1995-97 blood serum collected (outcome, measured one point in time)
 - o 2019-22 serum metabolic profiling
 - o “lifestyle variables were collected up to 18 years prior to breast cancer incidence” (line 411)
 - o Does the year of collection overlap with the serum collection (outcome measure)? Or is there a timing difference? There should be multiple lifestyle variable measures, pre and post serum collection? Do they only include one survey? Do they take averages of values?
- Women with current, regular menstruation used as a proxy for menopause status.
 - o (line28) “hormonal changes associated with menopause”
 - o (line 376) “we compared postmenopausal to pre-menopausal”
 - o However, the variable used is not menopause status, but regular menstruation. While in an overall sense, this may approximate, but other reasons for non-regular menstruation are hormonal disorder, birth control use, hormonal medication use, surgery (i.e. hysterectomy) and also peri menopausal.
 - o Birth control only an “ever” measure not current. While they state it isn’t “widely” used at 1984 study point, if they recruited over a 13 year period (again timing unclear), then that changes.
- Clustering analysis –
 - o (line 509) age is noted as a “major confounding factor” yet they do nothing to account for it (no stratification, no adjustment, no sensitivity analysis).
 - o What are the implications of leaving a key variable out of the cluster analysis? (would have liked to see even run as a sensitivity analysis as)
 - o Two justifications for exclusion: (1) worried it would only cluster based on age, (2) age is non-modifiable.
 - Yet they include reproductive items (like regular menstruation) which is also not modifiable?

Results

- Figure 1 I find very confusing. It is unclear the use of number ‘x’ number until you read the methods. The figure should be able to stand alone
- I question their reporting of the key finding:
 - o “suggesting accelerated metabolic aging with obesity” (line 19 – abstract)
 - o “thus we conclude that obesity accelerates metabolic aging” (line 407)
 - o Again “conclude” implies causation here when we don’t have this
 - o “obesity is associated with accelerated metabolic aging.” (line 446)
 - o This is based off of the cluster analysis, where general group characteristics are one group on average older and one group on average higher bmi (no significant differences of these variables across clusters reported). Why not look across strata of age and bmi? Especially because age (a confounder) is not accounted for in any way.
- Numerical data is not reported with confidence intervals or p-values. However, the Methods states they test significant differences across clusters (line 515), but they don’t report any in the body of the text. This should be added.
 - o They use terms such as “much wider” (239) and “a strong decrease” (170) but this appears to solely be visual analysis off of the figures. Please clarify if these you are just describing visual trends.
 - o They mention “low incidence of breast cancer in cluster 3 relative to cluster 2 is surprising” but is the 4% difference significant? There are different N between clusters (553 vs 1236)

Discussion

- Limitations could include no measurement of metabolic disorders/health conditions (noted in the background as influencing variables). Also, depending on the timing of lifestyle variables, the temporal distance between exposure (lifestyle variables) and outcome (serum metabolites) should be mentioned.

Minor comments:

In the abstract revise ‘leading to increased level...’ which infers causation that cannot be determined given the methodology

and design

Line 29 – revise education length to education or education level

Line 87 – revise to data cleaning

Clarify why variables with more than 30% missing were omitted versus multiple imputation or other imputation techniques

Line 207 – revise high age to older age

Line 397 – space and comma missing, should be 'this, the'

Table 1 – include "Cluster" header above "1", "2", "3" columns for clarity.

Table 1 – include breast cancer incidence as a row value.

Table 1 – include first two sentences of table notes in the title description, as unclear.

Table 2 – include RSQ in list of abbreviations.

Figure 5 – label c-e-g as c-d-e (down rather than across) as this is how they are grouped and described.

Version 1:

Reviewer comments:

Reviewer #1

(Remarks to the Author)

I appreciate the authors' efforts in addressing the previous comments. The revised manuscript has improved significantly; however, there are several new issues that need clarification:

1. Confusion Between Concepts: The manuscript appears to conflate "breast cancer-related lifestyle factors" with general "lifestyle factors." It is essential to clearly differentiate between these two concepts. Specifically, please identify which factors are categorized as breast cancer-related lifestyle factors and which are considered breast cancer-unrelated lifestyle factors.

2. Lack of Figure and Table References in Discussion: In the Discussion section, numerous results are mentioned without specifying their corresponding Figures or Tables. This omission makes it difficult to follow the results being discussed. Please include references to the relevant Figures and Tables for all results mentioned in this section.

3. Relevance of the Title: The manuscript's Results and Discussion sections rarely address associations with breast cancer risk. The analysis focuses predominantly on serum samples from breast cancer patients without explicitly linking findings to breast cancer risk factors. Therefore, the title "associations with breast cancer risk factors" may not accurately reflect the content of the paper. Consider revising the title to better match the scope of the study or expanding the analysis to directly address breast cancer risk factors.

Additionally, there are some formatting issues and typographical errors:

1. Table Layout: Table 1 could be presented in a horizontal layout to enhance the clarity of the information for each item.

2. Typographical Error: On Line 339, the term "weekly" should be corrected to "weakly."

Reviewer #2

(Remarks to the Author)

The revised manuscript is clearer and addresses most of the previous concerns. There are a few minor concerns that remain as detailed below.

Missing Data

It remains unclear why missing data beyond 30% missingness is considered 'missing not at random', while data below this threshold is considered 'missing at random' and therefore imputations are used. I would recommend missing data explanations to be included in the manuscript. For example, in the variables subsection in the Methods, include that a conservative threshold of 30% missingness was chosen to ensure robustness (cite), and that variables with greater than 30% missing values were related to disease and medication usage and therefore considered missing not at random. Then in the statistical analysis subsection in the Methods, when discussing imputations of missing data, include a statement about why those variables are considered 'missing at random'.

Minor notes:

In Figure 1, would recommend v instead of p , as p is typically assumed to mean significance level and isn't immediately clear.

Methods section: "Participants with diagnosed breast cancer" paragraph got separated from the Study Population section.

Methods section: "year" repeated twice when elaborating on different HUNT studies.

Sources of variation in the serum metabolome of female participants of the HUNT2 study – associations with breast cancer risk factors

Reviewers' comments:

Reviewer 1:

1. Sample Preprocessing and Batch Effects:

This work involves a considerable number of biological samples, potentially introducing batch effects in both NMR and MS spectra data. The "Method" section in the manuscript provides limited information regarding sample preprocessing. We suggest the authors to supplement this section with more detailed information on sample preprocessing steps, particularly focusing on how batch effects were mitigated. Authors should provide specific procedures for sample handling, including but not limited to sample collection, storage, preparation, and analysis, along with detailing how batch effects were addressed, such as employing quality control measures or correction methods.

Response: We agree that information regarding sample preprocessing is too limited. Therefore, we have now extended the method section with the following:

"NMR measurements:

In short, lipoprotein subfractions were quantified using the fully automated Bruker IVDr Lipoprotein Subclass Analysis (B.I.LISA™) panel, from Bruker BioSpin. The Bruker IVDr Lipoprotein Subclass Analysis (B.I.LISA™) panel was used to quantify lipoprotein subfractions. It reports lipid concentrations in total serum and in four main lipoprotein classes (VLDL, IDL, LDL, HDL) and 15 subclasses. It also provides serum levels of apolipoproteins (Apo-A1, Apo-A2, Apo-B), 12 calculated parameters (ratios of LDL-CH/HDL-CH and Apo-B/Apo-A1), and particle numbers of total serum, VLDL, IDL, LDL and LDL 1–6, totaling 112 lipoprotein subfractions. However, due to contamination in the serum samples, some subfractions, mostly from LDL-2 and LDL-4 particles, were excluded from further analysis, along with calculated parameters and particle numbers, leaving 89 lipoprotein subfractions. Small-molecular metabolites were quantified by integrating respective areas under metabolite peaks in CMPG spectra, after careful spectra preprocessing. The peaks were adjusted for T2 relaxation times. The NMR experiments were performed at two different labs, and although the same protocol was followed, a slight batch effect in the lipid profiles was observed. Since this effect was negligible and non-systematic, it was not accounted for in this study. Importantly, samples were run in a completely randomized order across the two labs. The proportion of variance introduced by the NMR measurements has been assessed through coefficients of variation (CVs) based on 46 quality control samples prepared from pooled serum samples of healthy controls, as previously described. CVs of the metabolites were below 15% and below 20% for 23 and 26 of the metabolites, respectively, and below 15% and 20% for 65 and 85, respectively, of the lipoprotein subfraction measurements.

MS data:

Quantitative analysis of more than 400 metabolites was performed using the Absolute IDQ p400 HR kit (96-well plates; Biocrates Life Sciences AG, Innsbruck, Austria). The kit combined direct flow injection for eight groups of lipids (phospholipids, glycerides, cholesterol esters or ceramides) and liquid chromatography (LC) high-resolution mass spectrometry (HRMS) for amino acids, biogenic amines, and a sum of hexoses.

Sample preparation for the MS measurements began with random placement of samples on measurement plates.²³ Subsequently, according to the protocol provided by the manufacturer, metabolites were extracted from 10 ml of blood serum (after applying the sample to a plate containing isotope-labeled internal standards and performing the on-plate chemical derivatization).

Sources of variation in the serum metabolome of female participants of the HUNT2 study – associations with breast cancer risk factors

Depending on the compounds analyzed, half of the extract was diluted in dedicated solutions. The metabolites were then analyzed using a system consisting of an Orbitrap Q Exactive Plus spectrometer (Thermo Fisher Scientific, Waltham, MA, USA) and a 1290 Infinity UHPLC liquid chromatograph (Agilent Santa Clara, USA), controlled by Xcalibur 4.1 software.

The obtained spectra and chromatograms were processed using Xcalibur 4.1 and MetIDQ DB110-2976 software (Biocrates Life Sciences, Innsbruck, Austria) dedicated by the kit manufacturer, obtaining concentration values in μM of individual metabolites present in specific samples.

After the array of metabolites concentrations was obtained, data analysis began with the detection and imputation of missing values. Metabolites for which measurements were missing completely at random (MCAR) in more or equal to 10% of samples were excluded from the dataset. Metabolites for which measurements were missed not at random (MNAR, i.e., values below the detection limit) in more or equal to 50% of samples were excluded from analysis. A threshold of 50% was adopted for MNAR values according to the recommendations of Chen and coworkers.⁵³ The remaining compounds ($n=284$) were logarithmically transformed and then the batch effect was corrected using an empirical Bayes method, assuming that samples measured using a single 96-well sample preparation plate represent one batch.⁵⁴ In the next step, the MNAR values were imputed with random numbers from the normal distribution truncated to the range from 0 to the median of the LOD values determined for all measurement plates. In turn, MCAR values were imputed using the k -nearest neighbors method (the neighborhood ranking was determined within each group of samples separately, and the missing values were imputed with the average value of the level of a given metabolite for the 3 most similar samples). After removing metabolites with more than 50% missing values, the CVs of inter- and intra-batch QC measurements was calculated according to a method given by Zhang et al.⁵⁵ Overall, the CV for 280 metabolites used in the quantitative analyses was below 15%. The median intra-batch coefficient of variation calculated for all metabolites in all QC samples ranged from 8-16%, and the median inter-batch CV was 18.73%.”.

2. Impact of Sample Storage Time:

The long timespan of sample collection in this study raises concerns regarding the impact of sample storage time, which is a critical factor influencing metabolite concentrations. Did the authors consider the effect of sample storage time during data preprocessing? If so, we recommend authors to elaborate on the methods used to exclude or correct for this factor in the Methods section. If not considered, authors should discuss this limitation in the Discussion section and explore its potential implications on the results.

Response: We agree that the extended sample storage is a limitation of our study. Although storage at -80°C or below is the recommended procedure for biofluids [Vaught J.B. Blood Collection, Shipment, Processing, and Storage. *Cancer Epidem Biomarkers Prev.* 2006;15(9):1582-1584], changes in the metabolite concentration during this process may be expected. Systematic analysis of this effect was beyond the scope of our study because we do not have access to the relevant set of control samples (nor to the set of fresh samples), thus we have to rely on previous studies. Studies on the effects of long-term storage are limited, and we only found studies on plasma samples. Nevertheless, such studies revealed some time-dependent effects on the concentration of plasma metabolites. For example, the concentration of plasma amino acids increased during the five years of storage (approximately 15%) [Haid et al. Long-Term Stability of Human Plasma Metabolites during Storage at -80°C . *J. Proteome Res.* 2018;17(1):203–211]. On the other hand, only a marginal effect on plasma metabolome composition up to 7 years of storage was reported in another study [Wagner-Golbs et al. Effects of Long-Term Storage at -80°C on the Human Plasma Metabolome. *Metabolites.* 2019;9(5):99].

Sources of variation in the serum metabolome of female participants of the HUNT2 study – associations with breast cancer risk factors

Hypothetical effects of the long-term storage on the composition of the serum metabolome have been included in the Discussion section:

“Moreover, the samples were collected over a period of two years and differences in sample handling procedures, such as differences in time from sample collection to centrifugation cannot be ruled out. It has previously been shown that a delay in centrifugation alters the levels of some metabolites,¹⁶ and that storage of plasma for up to 7 years has negligible impact on the metabolic profile,⁴⁹ while long-time storage may affect the levels of some lipids, amino-acids, and hexoses.⁵⁰ Importantly, these effects are randomly distributed over the samples and are commonly present for biobank samples.”

3. Features and Quantitative Information in NMR Data:

Figure 1 indicates the presence of 112+28 features in the NMR data. Please provide more information about these 112 features, including the compounds or metabolites they represent, and describe how their quantitative information was obtained. Additionally, do the 28 features represent small molecule metabolites? If so, please explain the identification and quantification methods for these metabolites.

Response: We agree that more information about these features is necessary and has been added to the methods section, as described in our answer to comment 1:

“In short, lipoprotein subfractions were quantified using the fully automated Bruker IVDr Lipoprotein Subclass Analysis (B.I.LISA™) panel, from Bruker BioSpin. The Bruker IVDr Lipoprotein Subclass Analysis (B.I.LISA™) panel was used to quantify lipoprotein subfractions. It reports lipid concentrations in total serum and in four main lipoprotein classes (VLDL, IDL, LDL, HDL) and 15 subclasses. It also provides serum levels of apolipoproteins (Apo-A1, Apo-A2, Apo-B), 12 calculated parameters (ratios of LDL-CH/HDL-CH and Apo-B/Apo-A1), and particle numbers of total serum, VLDL, IDL, LDL and LDL 1–6, totaling 112 lipoprotein subfractions. However, due to contamination in the serum samples, some subfractions, mostly from LDL-2 and LDL-4 particles, were excluded from further analysis, along with calculated parameters and particle numbers, leaving 89 lipoprotein subfractions. Small-molecular metabolites were quantified by integrating respective areas under metabolite peaks in CMPG spectra, after careful spectra preprocessing. The peaks were adjusted for T2 relaxation times.”

4. Consistency between NMR and MS Measured Metabolites:

The statement in the third paragraph of the Discussion section, "Findings between NMR- and MS-measured metabolites were consistent, confirming the validity of the molecular measurements," should be supported with more detailed comparative results. We suggest the authors to list the metabolites measured by both NMR and MS techniques, comparing their concentrations or expression levels, and discussing their correlation or consistency across the two methods.

Response: Thank you for pointing this out. These results have been presented in our previous publication [Mrowiec et al. Association of serum metabolome profile with the risk of breast cancer in participants of the HUNT2 study. *Frontiers in Oncology*. 2023; Vol.13] where correlations between all NMR-assessed lipid particles and MS-assessed lipids species are depicted, showing strong positive correlations between the majority of serum glycerides and concentrations of VLDLs, IDLs, and triglyceride-containing HDLs. An in-depth comparison between the two analytical platforms is however not within the scope of this study, thus this sentence has been removed.

5. Repetitive Measurement Experiments and Variations Introduced by Measurement Process:

Sources of variation in the serum metabolome of female participants of the HUNT2 study – associations with breast cancer risk factors

Variance in the data may include variations introduced by the measurement process. To assess this impact, we recommend authors to conduct some repetitive measurement experiments and provide an analysis of the obtained data in the Results section. Authors can estimate the proportion of variance introduced by the measurement process and discuss its implications for data interpretation and result reliability.

Response: The proportion of variance introduced by the NMR measurements has been previously assessed and described for our dataset [Debik et al. Lipoprotein and metabolite associations to breast cancer risk in the HUNT2 study. *British Journal of Cancer*. 2022; 127(8): 1515-1524]. The coefficients of variation (CVs) of the metabolites were below 15% and below 20% for 23 and 26 of the metabolites, respectively, based on 46 quality control samples prepared from pooled serum samples of healthy controls. Similarly, for the lipoprotein subfractions the CVs were below 15% and 20% for 65 and 85 of the variables, respectively. The following information has now been added into the methods section:

“The proportion of variance introduced by the NMR measurements has been assessed through coefficients of variation (CVs) based on 46 quality control samples prepared from pooled serum samples of healthy controls. CVs of the metabolites were below 15% and below 20% for 23 and 26 of the metabolites, respectively, and below 15% and 20% for 65 and 85, respectively, of the lipoprotein subfraction measurements.”

For MS measurements analysis of the quality of the absolute concentration of metabolites during the whole measurement were performed. After removing metabolites with more than 50% missing values, the CVs of inter- and intra-batch QC measurements was calculated according to a method given by Zhang et al. 2020 [Zhang et al. Five easy metrics of data quality for LC–MS-based global metabolomics. *Anal Chem*. 2020;92:12925-12933]. According to recommendations in the literature, in these types of experiments, measurements for 68% of the compounds tested should be considered acceptable (i.e., CV<30%). In our case, the CV for 280 metabolites used in the quantitative analyses was below 15%, which followed the current recommendations of the US Food and Drug Administration for data obtained in clinical trials [Food and Drug Administration. M10 Bioanalytical method validation and study sample analysis. Rockville, 2022]. The median intra-batch coefficient of variation calculated for all metabolites in all QC samples ranged from 8-16%, and the median inter-batch CV was 18.73%. Both median values (within and between plates) were similar to the results provided in reports of other authors [e.g., Moore et al. A metabolomics analysis of body mass index and postmenopausal breast cancer risk. *J Natl Cancer Inst*. 2018;110: 588-597; Playdon et al. Nutritional metabolomics and breast cancer risk in a prospective study. *Am J Clin Nutrition* 2017;106: 637-649). Figure 1 illustrate the inter-and intra-batch CVs.

Sources of variation in the serum metabolome of female participants of the HUNT2 study – associations with breast cancer risk factors

Figure 1 Inter- and intra-batch CVs for MS measurements

The following text has been added to the methods:

“After removing metabolites with more than 50% missing values, the CVs of inter- and intra-batch QC measurements was calculated according to a method given by Zhang et al.⁵⁵ Overall, the CV for 280 metabolites used in the quantitative analyses was below 15%. The median intra-batch coefficient of variation calculated for all metabolites in all QC samples ranged from 8-16%, and the median inter-batch CV was 18.73%.”

Reviewer 2:

Comments:

Debik et al present an overview of serum metabolic profiles of 2,283 females in the HUNT study in association with lifestyle factors.

The manuscript is overall, well written and within scope of the journal. The manuscript builds on existing literature to understand whether metabolomics can provide insight into risk factors for breast cancer. While not novel per se, the study does have several strengths and provides confirmatory as well as novel insights on specific metabolites.

Although, as mentioned above, the manuscript was well written, the abstract was difficult to follow. For example, the conclusion speaks to cluster analysis comparing obese and older individuals, but this was not outlined as an objective or in the results. Indeed, the aims are not presented, nor is the relevance to breast cancer articulated.

Response: Thank you for your valuable comment. The abstract has been rewritten to clarify the aim and main results of our study, and to make the link to breast cancer more clear.

“The aim of this study was to explore the intricate relationship between serum metabolomics and breast cancer-related lifestyle factors, shedding light on their impact on health in the context of breast cancer risk. Detailed metabolic profiles of 2283 female participants in the Trøndelag Health Study (HUNT study) were obtained through nuclear magnetic resonance (NMR) spectroscopy and mass spectrometry (MS).

We show that lifestyle-related variables can explain up to 30% of the variance in individual metabolites. Age and obesity were the primary factors affecting the serum metabolic profile, both associated with increased levels of triglyceride-rich very low-density lipoproteins (VLDL) and intermediate-density lipoproteins (IDL), amino acids and glycolysis-related metabolites, and

Sources of variation in the serum metabolome of female participants of the HUNT2 study – associations with breast cancer risk factors

decreased levels of high-density lipoproteins (HDL). Moreover, factors like hormonal changes associated with *menstruation* and contraceptive use or education *level* influence the metabolite levels.

Participants were clustered into three distinct clusters based on lifestyle-related factors, revealing metabolic similarities between obese and older individuals, despite diverse lifestyle factors, suggesting accelerated metabolic aging with obesity. Our results show that metabolic associations to cancer risk may partly be explained by modifiable lifestyle factors.”

Although I acknowledge that lifestyle is a term widely used in public/population health (I also used to use it), I would encourage the authors to reflect on the use of the term, and perhaps provide alternative wording. Suggested reading: [https://ukpublichealthnetwork.org.uk/lifestyle-a-plea-to-abandon-the-use-of-this-word-in-public-health/#:~:text=“Lifestyle” is a loaded term, responsible for their own health.](https://ukpublichealthnetwork.org.uk/lifestyle-a-plea-to-abandon-the-use-of-this-word-in-public-health/#:~:text=“Lifestyle”%20is%20a%20loaded%20term, responsible%20for%20their%20own%20health.)

Response: We agree that the term lifestyle may be a loaded term and is somewhat unprecise, as the term lifestyle related factors in our study refers collectively to demographics, clinical measurements and socio-economic factors in addition to adjustable lifestyle factors. However, we have clarified this in the results: *“60 variables related to demography, lifestyle, and socio-economic factors, and anthropometric and clinical measurements (lifestyle-related factors in short) were retained for statistical analyses”* and we argue that this term is the best compromise between precise language and making the manuscript easy to read.

Additional comments by section are outlined below.

Methods

- The timing of lifestyle variable collection (exposure) is unclear and possibly problematic.
 - o 1984 start of cohort
 - ♣ “repeat measures every 10 years” (455)
 - ♣ but also “4 surveys collected between 1984 and 2019” (529)?
 - o 22 year follow up time
 - ♣ 1984 start would bring us to 2006
 - ♣ But if individuals were recruited over a 13 year period, then could get to 2019 end collection time.
 - o 1995-97 blood serum collected (outcome, measured one point in time)
 - o 2019-22 serum metabolic profiling
 - o “lifestyle variables were collected up to 18 years prior to breast cancer incidence” (line 411)
 - o Does the year of collection overlap with the serum collection (outcome measure)? Or is there a timing difference? There should be multiple lifestyle variable measures, pre and post serum collection? Do they only include one survey? Do they take averages of values?

Response: Thank you for bringing this unclarity to our notice. The HUNT study is a longitudinal population-based study, where biological material, questionnaire data, and clinical measurements have been collected over four waves: HUNT1 (1984–1986), HUNT2 (1995–1997), HUNT3 (2006–2008) and HUNT4 (2017–2019). Our study is based solely on the HUNT2 study. All lifestyle variables included have been collected at the time of blood sampling at HUNT2. Serum samples were stored at the biobank until metabolomic profiling in 2019–22. Individuals were matched with the Norwegian

Sources of variation in the serum metabolome of female participants of the HUNT2 study – associations with breast cancer risk factors

Cancer Registry in 2019 to identify all participants from the HUNT2 study that developed breast cancer between data collection (1995-97) and follow-up in 2019. However, for this study, we removed all women with a breast cancer diagnosis within three years after blood sampling and treated all participants as breast-cancer free at the time of sampling.

The following has been added to the methods part to clarify:

“The HUNT study is a prospective, longitudinal, population-based health study, conducted in Trøndelag, Mid-Norway since 1984, with repeated measures every tenth year: HUNT1 (1984-86), HUNT2 (1995-97), HUNT3 (2006-08) and HUNT4 (2017-19). It includes self-reported questionnaire-based health data, clinical examinations, and biological sampling from over 100,000 individuals.

In the study by Debik, et al. all HUNT2 female participants with a later breast cancer diagnosis within a 22-year follow-up period were identified (n = 1208), by matching HUNT2 individuals with the Norwegian Cancer Registry in 2019.”

- Women with current, regular menstruation used as a proxy for menopause status.

o (line28) “hormonal changes associated with menopause”

o (line 376) “we compared postmenopausal to pre-menopausal”

o However, the variable used is not menopause status, but regular menstruation. While in an overall sense, this may approximate, but other reasons for non-regular menstruation are hormonal disorder, birth control use, hormonal medication use, surgery (i.e. hysterectomy) and also peri menopausal.

Response: The sentence in line 376 refers to our previous study. In the current study, we do not use menopausal status at all because of the uncertainty related to the menopausal status in our cohort, due to a large missingness (approx. 80%). The mean age of the study population is 51 years, suggesting that many postmenopausal women failed to fill in their menopausal age. We agree that irregular menstruation does not imply menopause, and therefore, throughout the manuscript, we do not use the term menopausal status to describe our cohort. We have reformulated the abstract so that the term menopause is replaced with menstruation, which more accurate:

*“Moreover, factors like hormonal changes associated with **menstruation** and contraceptive use or education **level** influence the metabolite levels.”*

In the discussion we state the following, to make it clear that we are not implying menopause:

*“In the present study we observe lower levels of lipoproteins **among participants who reported current regular menstruation.**”*

♣ Birth control only an “ever” measure not current. While they state it isn’t “widely” used at 1984 study point, if they recruited over a 13 year period (again timing unclear), then that changes.

Response: Thank you for your comment. We did in fact have a variable for current use of birth control pills (see Table 1). As mentioned above, the collection of lifestyle variables (including the use of birth control pills, current use or ever use) occurred in the years 1995-97.

- Clustering analysis –

o (line 509) age is noted as a “major confounding factor” yet they do nothing to account for it (no stratification, no adjustment, no sensitivity analysis).

Sources of variation in the serum metabolome of female participants of the HUNT2 study – associations with breast cancer risk factors

- ♣ What are the implications of leaving a key variable out of the cluster analysis? (would have liked to see even run as a sensitivity analysis as)
- ♣ Two justifications for exclusion: (1) worried it would only cluster based on age, (2) age is non-modifiable.
- Yet they include reproductive items (like regular menstruation) which is also not modifiable?

Response: Thank you for your insightful comment. Age was not included into the cluster analysis because it would dominate the clustering. We have previously observed that age has a profound impact on the serum metabolic profile, please see [Mrowiec et al. Association of serum metabolome profile with the risk of breast cancer in participants of the HUNT2 study. *Frontiers in Oncology*. 2023; Vol.13]. This reference has been added to the manuscript. Age is however still represented indirectly, as it is highly correlated with several of the other included lifestyle variables such as body weight or blood pressure. However, the influence of age on the metabolic profile is depicted by the correlation analysis in Figure 3 and in Figure 4. In addition, the cluster-specific metabolic profiles are compared, which consist of different age groups of participants in Figure 6. Finally, age was included into Ridge regression.

Results

- Figure 1 I find very confusing. It is unclear the use of number 'x' number until you read the methods. The figure should be able to stand alone

Response: Thank you for pointing this out. The figure has been modified to describe the number of individuals and variables in each step.

- I question their reporting of the key finding:

o "suggesting accelerated metabolic aging with obesity" (line 19 – abstract)

o "thus we conclude that obesity accelerates metabolic aging" (line 407)

♣ Again "conclude" implies causation here when we don't have this

o "obesity is associated with accelerated metabolic aging." (line 446)

Response: Thank you for bringing this to our notice. We agree that our findings are explorative and thus hypothesis generating, and do not imply causation. We have now rephrased lines 407 and 446:

Line 407: "*thus we hypothesize that obesity accelerates metabolic aging*"

Line 446: "*obesity may accelerate metabolic aging*"

o This is based off of the cluster analysis, where general group characteristics are one group on average older and one group on average higher bmi (no significant differences of these variables across clusters reported). Why not look across strata of age and bmi? Especially because age (a confounder) is not accounted for in any way.

Response: Thank you for your comment. The rationale for applying cluster analysis to group the participants based on several factors was that most of the lifestyle variables are highly correlated with each other, thus looking at the influence of BMI on the serum metabolome alone, does not take into account the influence of age, and vice versa. As mentioned previously, age although not directly, is indirectly represented in the cluster analysis as it is highly correlated with the majority of the other lifestyle factors, including education length and reproductive factors. Age has been included in all remaining analysis and comparisons, including Ridge regression.

Sources of variation in the serum metabolome of female participants of the HUNT2 study – associations with breast cancer risk factors

- Numerical data is not reported with confidence intervals or p-values. However, the Methods states they test significant differences across clusters (line 515), but they don't report any in the body of the text. This should be added.

Response: Thank you for noticing. P-values have now been added to Table 1 so highlight which variables are significantly different across the clusters.

o They use terms such as “much wider” (239) and “a strong decrease” (170) but this appears to solely be visual analysis off of the figures. Please clarify if these you are just describing visual trends.

Response: Thank you for your comment. It is correct that these are visual observations. We argue that this should already be clear in the manuscript, as we in line 107 use the word “observed”, while in line 239 we use the phrase “show the distribution”, and we state that the observed difference is based on comparing the breadth of the peaks in the density plot.

We have made the following change in line 239 to make this even more clear:

*“For MS-measured metabolites, a strong decrease in glycine with increased WHR was **observed by visual inspection**, while it was quite constant over different ranges of BMI (Figures 4V and X).”*

o They mention “low incidence of breast cancer in cluster 3 relative to cluster 2 is surprising” but is the 4% difference significant? There are different N between clusters (553 vs 1236)

Response: Thank you pointing this out. We have now tested the significance of the differences in breast cancer incidence between the clusters through a Chi-squared test, which revealed that these differences are in fact not significant. The paragraph has been modified as follows:

*“The low incidence of breast cancer within cluster 3 compared to cluster 2 might be surprising, considering known breast cancer risk factors. **However, this difference is not statistically significant**. It is important to note that the time-to-diagnosis was much shorter for participants in clusters 1 and 3 than cluster 2, and that the lifestyle variables were collected up to 18 years prior to breast cancer incidence. In other words, this suggests that reaching a high age while preventing severe overweight, such as participants in cluster 3, reduces breast cancer risk. These participants have also reported the highest number of pregnancies and scored low on alcohol consumption and smoking, factors, which are associated with lower breast cancer risk.”*

Discussion

- Limitations could include no measurement of metabolic disorders/health conditions (noted in the background as influencing variables). Also, depending on the timing of lifestyle variables, the temporal distance between exposure (lifestyle variables) and outcome (serum metabolites) should be mentioned.

Response: Lifestyle variables were collected in parallel with blood draw, thus the temporal distance between exposure and outcome is the same for all lifestyle variables.

Minor comments:

In the abstract revise ‘leading to increased level...’ which infers causation that cannot be determined given the methodology and design

Sources of variation in the serum metabolome of female participants of the HUNT2 study – associations with breast cancer risk factors

Response: The abstract has been changed to emphasize that this is not a causation, but an association.

“We show that lifestyle-related variables can explain up to 30% of the variance in individual metabolites. Age and obesity were the primary factors affecting the serum metabolic profile, both associated with increased levels of triglyceride-rich very low-density lipoproteins (VLDL) and intermediate-density lipoproteins (IDL), amino acids and glycolysis-related metabolites, and decreased levels of high-density lipoproteins (HDL). Moreover, factors like hormonal changes associated with menstruation and contraceptive use or education level influence the metabolite levels.”

Line 29 – revise education length to education or education level

Response: Thank you for your comment. Education length has been changed to education level.

Line 87 – revise to data cleaning

Clarify why variables with more than 30% missing were omitted versus multiple imputation or other imputation techniques

Response: The decision to omit variables with more than 30% missing data was based on several considerations. Firstly, the threshold of 30% was chosen as a conservative measure to ensure the robustness of our analysis. Variables with high levels of missing data can introduce bias or inaccuracies into the results, especially if the missingness is not completely at random see [Sterne et al. Multiple imputation for missing data in epidemiological and clinical research: potential and pitfalls *BMJ*. 2009; Vol.338]. Secondly, imputation techniques, such as multiple imputation are indeed powerful tools for handling missing data, they also come with their own assumptions and limitations [Azur et al., Multiple imputation by chained equations: what is it and how does it work? *Int J Methods Psychiatr Res*. 2011]. For instance, multiple imputation assumes that the data are missing at random, which we believe is not the case for many of the variables with high percentage of missing variables in our dataset. The highest percentage of missing variables are variables related to diseases or medication usage.

Line 207 – revise high age to older age

Response: This has been corrected.

Line 397 – space and comma missing, should be ‘this, the’

Response: Thank you for noticing. This has been corrected.

Table 1 – include “Cluster” header above “1”, “2”, “3” columns for clarity.

Response: The table has now been modified.

Table 1 – include breast cancer incidence as a row value.

Response: Breast cancer incidence has now been included into the table.

Sources of variation in the serum metabolome of female participants of the HUNT2 study – associations with breast cancer risk factors

Table 1 – include first two sentences of table notes in the title description, as unclear.

Response: The two sentences have now been moved to the title of the table.

Table 2 – include RSQ in list of abbreviations.

Response: RSQ has been added to the list of abbreviations in Figure 2.

Figure 5 – label c-e-g as c-d-e (down rather than across) as this is how they are grouped and described.

Response: Thank you for pointing this out. The figure has now been modified.

Sources of variation in the serum metabolome of female participants of the HUNT2 study – associations with breast cancer risk factors

Reviewers' comments:

Reviewer 1:

1. Confusion Between Concepts: The manuscript appears to conflate "breast cancer-related lifestyle factors" with general "lifestyle factors." It is essential to clearly differentiate between these two concepts. Specifically, please identify which factors are categorized as breast cancer-related lifestyle factors and which are considered breast cancer-unrelated lifestyle factors.

Response: We agree that the manuscript would benefit from a clearer separation between lifestyle variables associated with breast cancer risk and remaining lifestyle variables. All variables included in the study were initially selected for a breast-cancer association study, however, not all variables have been repeatedly found to be associated with breast cancer risk. The introduction has now been modified accordingly:

"In this paper, we aimed to investigate the main sources of variation, considering both breast cancer-related and other lifestyle factors, in the female serum metabolome, offering valuable insight into metabolic pathways that might be relevant for breast cancer prevention. Breast cancer-related factors considered in this study are alcohol consumption, menarche age, height, age at first pregnancy, number of full-term pregnancies, obesity, physical activity, systemic menopausal estrogen use, and birth control pill use.²⁹"

2. Lack of Figure and Table References in Discussion: In the Discussion section, numerous results are mentioned without specifying their corresponding Figures or Tables. This omission makes it difficult to follow the results being discussed. Please include references to the relevant Figures and Tables for all results mentioned in this section.

Response: Thank you for pointing this out. References to figures and tables have now been included throughout the discussion section.

3. Relevance of the Title: The manuscript's Results and Discussion sections rarely address associations with breast cancer risk. The analysis focuses predominantly on serum samples from breast cancer patients without explicitly linking findings to breast cancer risk factors. Therefore, the title "associations with breast cancer risk factors" may not accurately reflect the content of the paper. Consider revising the title to better match the scope of the study or expanding the analysis to directly address breast cancer risk factors.

Response: We agree that breast cancer is not the main focus of this paper, and have therefore now removed the second part of the title, so that it is: "Sources of variation in the serum metabolome of female participants of the HUNT2 study"

Additionally, there are some formatting issues and typographical errors:

1. Table Layout: Table 1 could be presented in a horizontal layout to enhance the clarity of the information for each item.

Response: The table has now been modified.

2. Typographical Error: On Line 339, the term "weekly" should be corrected to "weakly."

Sources of variation in the serum metabolome of female participants of the HUNT2 study – associations with breast cancer risk factors

Response: Thank you for noticing. This has been corrected in the revised manuscript.

Reviewer 2:

Missing Data

It remains unclear why missing data beyond 30% missingness is considered 'missing not at random', while data below this threshold is considered 'missing at random' and therefore imputations are used. I would recommend missing data explanations to be included in the manuscript. For example, in the variables subsection in the Methods, include that a conservative threshold of 30% missingness was chosen to ensure robustness (cite), and that variables with greater than 30% missing values were related to disease and medication usage and therefore considered missing not at random. Then in the statistical analysis subsection in the Methods, when discussing imputations of missing data, include a statement about why those variables are considered 'missing at random'.

Response: Thank your feedback. We appreciate the opportunity to clarify our approach to handling missing data related to demography and lifestyle. A conservative threshold of 30% missingness was chosen to ensure a good balance between robustness while keeping as many variables as possible for data analysis. This threshold was selected based on our specific dataset, as variables with high missingness were predominantly related to disease and medication usage, which we considered to be 'missing not at random', while variables with low missingness were in general more in line with the assumption 'missing at random' (but not with a stringent 30% cut-off on the type of missingness). Please see the figure below, which shows degree of missingness for the original full dataset. The blue dashed line indicates the cut-off for 30% missingness, and clearly shows that it predominantly reduces the original data set by removing variables with a high degree of missingness while keeping variables with a relatively low degree of missingness.

The following has been added to the Methods section:

"A conservative threshold of 30% missingness was selected based on the characteristics of the available lifestyle variables in this study to ensure a good balance between robustness and retaining as many variables as possible for data analysis. Variables with high missingness were predominantly related to diseases and medication usage, which we considered to be 'missing not at random'. In contrast, variables with low missingness generally aligned with the assumption of being 'missing at random'.

Sources of variation in the serum metabolome of female participants of the HUNT2 study – associations with breast cancer risk factors

Sources of variation in the serum metabolome of female participants of the HUNT2 study – associations with breast cancer risk factors

Minor notes:

1. In Figure 1, would recommend v instead of p, as p is typically assumed to mean significance level and isn't immediately clear.

Response: This has now been modified.

2. Methods section: "Participants with diagnosed breast cancer" paragraph got separated from the Study Population section.

Response: Thank you for pointing this out. This has now been corrected.

3. Methods section: "year" repeated twice when elaborating on different HUNT studies.

Response: This has now been corrected in the revised version.